# Collapsed Effective Operators for Higher-order Structures

**Maximilian Krahn** [* 1]   **Lennart Bastian** [* 1 2 3]   **Vikas Garg** [† 4 5]   **Björn Schuller** [† 1 2 3]   **Tolga Birdal** [† 1]

## Abstract

Higher-order structures are powerful relational modeling tools, yet existing spectral operators decompose topology into separate ranks, leaving practitioners to fuse information back to vertices through ad-hoc choices. We introduce *Collapsed Effective Operators*, which condense higher-order degrees of freedom into a single vertex-level operator via Schur complementation of a graded Laplacian. This yields a (generally dense) operator that encodes long-range interactions mediated by topology and is applicable to arbitrary higher-order constructs. We show it preserves positive semi-definiteness with a strict spectral upper bound relative to the rank-0 Hodge Laplacian, effectively lowering system energy under higher-order connectivity. Empirically, our operator improves spectral clustering, signal smoothing and enables the inclusion of topological features in neural network architectures via positional encoding. The code is available here.

## 1. Introduction

The collective behavior of complex systems, from protein complexes and chemical reactions, to neural circuits and collaborative networks, is fundamentally *polyadic*: it emerges from the simultaneous interaction of multiple components (Battiston et al., 2020; Bick et al., 2023). While graphs have emerged as a universal language for modeling relational data, they are fundamentally limited through *dyadic* edges and cannot represent higher-order interactions directly. This insight has motivated advances in algorithms that operate on higher-order constructs which explicitly in-corporate polyadic relations, such as hypergraphs, simplicial complexes, and combinatorial complexes (Gao et al., 2020; Barbarossa & Sardelletti, 2020; Hajij et al., 2022) (cf. Fig. 2).

Spectral graph theory connects the algebraic properties of the graph Laplacian to geometric and topological characteristics of the underlying structure (Chung, 1997). The eigenvalues and eigenvectors of the Laplacian govern diffusion processes, define notions of smoothness, and enable tools for clustering, filtering, and representation learning (Shuman et al., 2013). This perspective underpins Graph Neural Networks, which propagate information along edges to learn node and graph-level representations (Kipf & Welling, 2017; Veličković et al., 2018). Analogous spectral theory on higher-order structures has been developed within algebraic topology and topological signal processing, building on the Hodge Laplacian (Eckmann, 1944; Lim, 2020) and its variants/extensions (Barbarossa & Sardelletti, 2020; Schaub et al., 2020; Viganò et al., 2026), and has recently been incorporated into neural architectures, spawning the field of Topological Deep Learning (Bodnar et al., 2021a; Hajij et al., 2022; Papamarkou et al., 2024). Yet across all of these settings, leveraging higher-order spectral information for node-level tasks has proven challenging.

Despite the expressive power of imposing higher-order relationships on data, the signal of interest in most applications **lives on a specific, single rank**. For instance, node classification (Sen et al., 2008; Yang et al., 2016), molecular property prediction (Gilmer et al., 2017; Schütt et al., 2017), and spectral clustering (Shi & Malik, 2000) all require vertex-level predictions. Yet topological information encoded in higher-order cells does not directly transfer to vertices, creating a persistent *fusion problem*: how should rank-specific representations be combined to inform node-level predictions? Classical approaches to hypergraph clustering (Zhou et al., 2006) and motif-based analysis (Benson et al., 2016) sidestep this by operating on higher-order cells and then projecting or compressing the results onto the vertex set using application-specific heuristics. Higher-order message-passing (HOMP) methods take a different route, propagating features from nodes to higher-order cells and back (Bodnar et al., 2021a; Hajij et al., 2022). In both paradigms, fusing multi-rank information into a coherent vertex-level signal requires domain knowledge and architectural decisions that may not generalize across tasks (Papillon et al., 2025).

---

[*]Equal Contribution.   [†]Equal senior-authorship. [1]Department of Computing, Imperial College London, London, UK [2]Chair of Health Informatics, Technical University of Munich, Germany [3]Munich Center for Machine Learning, Germany [4]Aalto University, Finland [5]YaiYai Ltd. Correspondence to: Maximilian Krahn <max.krahn22@imperial.ac.uk>, Lennart Bastian <l.bastian@imperial.ac.uk>.

*Proceedings of the 43rd International Conference on Machine Learning*, Seoul, South Korea. PMLR 306, 2026. Copyright 2026 by the author(s).

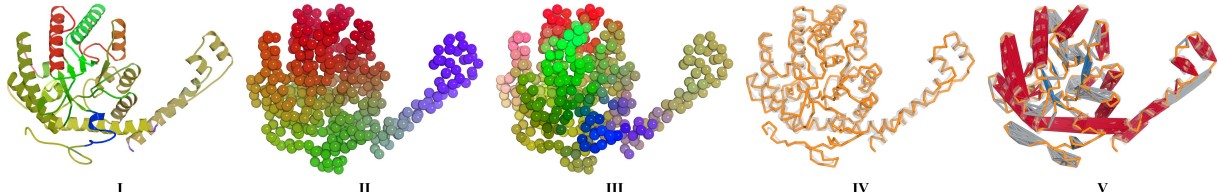

**Figure 1.** Our *Collapsed effective operators* reveal secondary-structure-aware spectral modes. On protein 1A0C (437 residues) from Topotein (Wang et al., 2025), we color residues by the leading spectral embedding (top-3 joint-PCA components of the leading eigenvectors) of our collapsed operator **S** versus the graph Laplacian $\mathbf{L}^G$, aligned so matching modes share colors. **Left to right:** (I) the secondary structure, shown as helical ribbons and $\beta$-strand arrows; (II) $\mathbf{L}^G$ produces smoother, backbone-sequential modes; (III) modes from **S** are organized around secondary-structure elements (SSEs); (IV) the backbone graph underlying $\mathbf{L}^G$; and (V) **S** as a vertex-level operator, with SSE cells collapsed into within-cell couplings on the backbone and colored by class (helix red, sheet blue, coil grey).

Several spectral operators for higher-order structures have been conceived. The Hodge Laplacian (Eckmann, 1944; Lim, 2020) generalizes the graph Laplacian to $k$-dimensional cells, producing a separate operator $\Delta_k$ at each rank that couples adjacent ranks $k \leftrightarrow k \pm 1$. While mathematically elegant, this rank-wise decomposition creates a practical bottleneck: fusing information across ranks back to the vertex level remains application-dependent, often requiring **handcrafted aggregation schemes or domain-specific heuristics**. Extensions such as the Dirac operator (Calmon et al., 2023), persistent Laplacians (Mémoli et al., 2022), and the multi-order Laplacian (Lucas et al., 2020) address various limitations, yet all share a common shortcoming: they either operate rank-by-rank or aggregate ranks additively without accounting for how higher-order cells mediate vertex dynamics. None yields a node-level operator that encodes the complete influence of the higher-order structure through a single, principled construction.

To address this gap, we propose *Collapsed Effective Operators*. Our approach is inspired by the physics of effective theories, where complex systems are simplified **by integrating out degrees of freedom that are not directly observable** or of primary interest (Bell & Wilson, 1974; Feshbach, 1958). We apply this philosophy to signals on topological structures: rather than explicitly computing dynamics across all cell ranks, we collapse the higher-order structure onto the vertex set via the Schur complement (Boyd & Vandenberghe, 2004), yielding an operator that acts solely on nodes (see fig. 1). Unlike Laplacians, this operator may be locally dense, reflecting non-local couplings induced by higher-order topology, and its structure depends highly on the specific topological choices made in constructing the complex. However, we demonstrate that this is not a limitation but a feature: the effective operator encodes **how higher-order modeling assumptions influence vertex-level dynamics.** In summary, we **contribute**:

- We derive an *effective operator* on vertices that captures the influence of higher-order structures
- We show that our devised operator is positive semidefinite and admits a clear spectral interpretation, reducing the

effective conductance in nodes that share common cells.
- Our operator **overcomes rank-constrained limitations**, aggregating topological features such as isolated cells down to the node-level, which existing Laplacians cannot easily model. We provide an algorithm to efficiently compute this operation, using regularized solvers that leverage the sparse block structure in the lifted space.
- We **validate our method across diverse tasks**, showing that it acts as a topologically selective filter that smooths noise while preserving higher-order structure, improves spectral clustering, recovers interleaved protein secondary structures invisible to geometric methods, and achieves compelling accuracy on molecular benchmarks.

## 2. Background and Motivation

We review the necessary background on graphs and higher-order complexes, highlighting the limitations of rank-constrained operators, such as higher-order Laplacians, to motivate the effective operator construction in sec. 3.

### 2.1. Graphs and Higher-Order Structures

**Notation.** We consider an undirected graph $G = (V, E)$ consisting of a finite set of $n$ vertices $V = \{v_1, \ldots, v_n\}$ and a set of $m$ edges $E \subseteq V \times V$. To represent the graph algebraically, we fix an arbitrary reference orientation on each edge yielding the signed incidence matrix $\mathbf{E} \in \{-1, 0, 1\}^{n \times m}$, defined by $E_{ij} = 1$ if vertex $v_i$ is the head (target) of edge $e_j$, $E_{ij} = -1$ if $v_i$ is the tail (source) of $e_j$, and 0 otherwise. For an undirected graph, the orientation is arbitrary. Let $\mathbf{L^G} = \mathbf{E}\mathbf{E}^\top$ denote the graph Laplacian, a positive semidefinite (PSD) operator on $G$.

Next, we introduce the cellular setting in which we formulate the main construction. We use CW complexes as the default construct; they are broad enough to allow flexible cells and attaching maps, while still containing simplicial complexes, which are broadly used across computational topology, geometry processing, discrete exterior calculus, mesh analysis, and topological deep learning (Wardetzky et al., 2007; Schaub et al., 2020; Bodnar et al., 2021a; Hajij

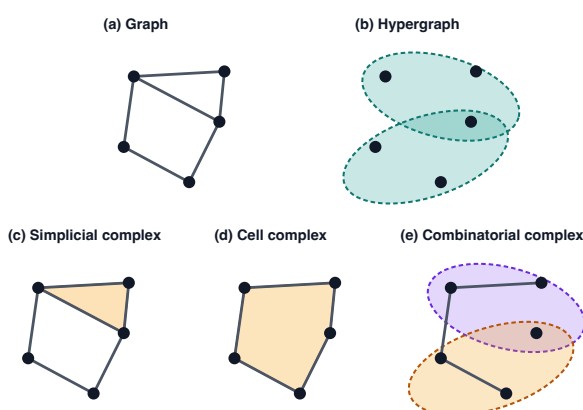

**Figure 2.** Topological domains of increasing flexibility. **(a)** *Graph*: pairwise edges. **(b)** *Hypergraph*: edges of arbitrary size, all flat (no rank). **(c)** *Simplicial complex*: higher-order simplices, each determined by its faces. **(d)** *Cell complex*: general cells whose boundaries need not be simplices. **(e)** *Combinatorial complex*: possibly overlapping cells with explicit rank.

et al., 2022). Fig. 2 contextualizes these structures within the broader field of topological domains.

**Definition 2.1** (CW Complex). A finite *CW complex X* is constructed inductively by attaching cells. Starting with a 0-skeleton $X^0$ (a finite set of vertices), the $k$-skeleton $X^k$ is formed from $X^{k-1}$ by attaching $k$-cells $e^k_\alpha$ via attaching maps $\varphi_\alpha : S^{k-1} \to X^{k-1}$, where $S^{k-1} = \partial D^k$ is the boundary sphere of a $k$-disk. The full complex is $X = \bigcup_{k=0}^K X^k$, and the *rank* or *grade* of a cell is its dimension $k$. Unlike simplicial complexes, CW complexes allow flexible attaching maps: a cell need not be a simplex, and a boundary can wrap around the lower skeleton non-locally or with multiplicity (Hatcher, 2002).

A simplicial complex is the special case where the cells are simplices and the attaching maps glue each simplex to its faces. Equivalently, it is a collection of non-empty finite subsets of a vertex set that is closed under taking non-empty subsets (Edelsbrunner & Harer, 2010, Ch. 3). Thus the CW-complex formulation below includes the standard simplicial setting without requiring separate notation.

**Definition 2.2** (Cellular Cochains and Grading). Let $X_k$ denote the finite set of $k$-cells of $X$, with $n_k = |X_k|$. The cellular chain space $C_k(X) \cong \mathbb{R}^{n_k}$ is the vector space spanned by oriented $k$-cells. The cellular cochain space

$$C^k(X) := \mathrm{Hom}(C_k(X), \mathbb{R}) \cong \mathbb{R}^{n_k} \tag{1}$$

is the space of real-valued signals on $k$-cells. The *graded cochain space* is the direct sum

$$C^\bullet(X) := \bigoplus_{k=0}^K C^k(X), \tag{2}$$

so a graded cochain $\mathbf{x} \in C^\bullet(X)$ decomposes as $\mathbf{x} = (\mathbf{x}_0, \dots, \mathbf{x}_K)$, where $\mathbf{x}_k \in C^k(X)$ is the rank-$k$ component. The grading is therefore the decomposition by cell dimension.

**Definition 2.3** (Boundary and Coboundary Maps). The cellular boundary map $\partial_k : C_k(X) \to C_{k-1}(X)$ records the signed incidence of each oriented $k$-cell with the $(k-1)$-cells in its boundary. Its matrix representation is $\mathbf{B}_k \in \mathbb{R}^{n_{k-1} \times n_k}$. The adjoint coboundary $d_{k-1} = \partial_k^* : C^{k-1}(X) \to C^k(X)$ is represented by $\mathbf{B}_k^\top$ under the standard inner product. Boundary maps satisfy

$$\partial_{k-1} \circ \partial_k = 0 \quad \text{for all } k, \tag{3}$$

signifying that "the boundary of a boundary is empty" (Hatcher, 2002, Lemma 2.1).

**Definition 2.4** (Hodge Laplacian). Let $X$ be a finite CW complex with boundary matrices $\mathbf{B}_k$. The $k$-th *Hodge Laplacian* is the rank-local operator

$$\mathbf{\Delta}_k = \underbrace{\mathbf{B}_k^\top \mathbf{B}_k}_{\mathbf{L}_k^{\mathrm{down}}} + \underbrace{\mathbf{B}_{k+1} \mathbf{B}_{k+1}^\top}_{\mathbf{L}_k^{\mathrm{up}}}, \tag{4}$$

where $\mathbf{L}_k^{\mathrm{down}}$ couples $k$-cells through shared $(k-1)$-faces and $\mathbf{L}_k^{\mathrm{up}}$ couples $k$-cells that are cofaces of a common $(k+1)$-cell (Eckmann, 1944).

The Hodge Laplacian is defined separately at each rank of the grading. This rank-locality is mathematically natural, but it leaves vertex-level tasks with the problem of aggregating information from several different cochain spaces.

### 2.2. Higher-order Operators

Just as the graph Laplacian enables spectral analysis and signal processing on graph vertices, Hodge Laplacians enable diffusion and signal processing on each rank of a CW complex. However, they remain rank-local: $\mathbf{\Delta}_k$ acts on $C^k(X)$, while node-level tasks require an operator on $C^0(X)$ that still reflects the influence of higher grades. The literature, therefore, lacks a unified operator that maintains the boundary-mediated structure of a complex while aggregating multi-rank information to a chosen rank. *Collapsed Effective Operators* address this gap, providing a mechanism for higher-order connections to mediate the flow of information at a specific rank, as we show next.

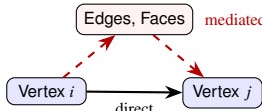

## 3. Collapsed Higher-Order Operators

We now outline our approach for *Collapsed Effective Operators*, which marginalize higher-order cells onto vertices via the Schur complement (Boyd & Vandenberghe, 2004) (see

Appendix A.1). This construction preserves positive semi-definiteness, admits a clear energy interpretation, and can be applied efficiently with controlled regularization error.

## 3.1. The Graded Laplacian

We formulate the construction on the graded cochain space $C^\bullet(X)$ of a finite CW complex $X$ (def. 2.2). Thus $C^k(X) \cong \mathbb{R}^{n_k}$ carries real-valued signals on $k$-cells and a graded signal decomposes as $\mathbf{x} = (\mathbf{x}_0, \ldots, \mathbf{x}_K)$ with $\mathbf{x}_k \in C^k(X)$. This grading is the natural setting for cross-rank operators and underlies the topological Dirac operator (Bianconi, 2021; Calmon et al., 2023), which couples adjacent ranks via boundary maps but is not reducible to the vertex space. Inspired by this perspective, we define an operator that couples all ranks simultaneously. Throughout the main exposition, $\mathbf{B}_k \in \mathbb{R}^{n_{k-1} \times n_k}$ denotes the signed cellular boundary matrix between ranks $k$ and $k-1$.

**Proposition 3.1** (Graded Laplacian). *Let $\{\mathbf{B}_k\}_{k=1}^K$ be boundary matrices between adjacent ranks, with adjacency weights $\beta_k \geq 0$ and coupling weights $\gamma_k \geq 0$. The* Graded Laplacian $\mathbf{L}^\star : C^\bullet \to C^\bullet$ *is the symmetric operator:*

$$
\mathbf{L}^\star = \begin{pmatrix}
\mathbf{D}_0 & -\gamma_0 \mathbf{B}_1 & & \\
-\gamma_0 \mathbf{B}_1^\top & \mathbf{D}_1 & \ddots & \\
& \ddots & \ddots & -\gamma_{K-1} \mathbf{B}_K \\
& & -\gamma_{K-1} \mathbf{B}_K^\top & \mathbf{D}_K
\end{pmatrix}
\tag{5}
$$

*where the diagonal blocks are weighted rank-local Laplacians:*

$$
\mathbf{D}_k = \beta_k \mathbf{B}_k^\top \mathbf{B}_k + \beta_{k+1} \mathbf{B}_{k+1} \mathbf{B}_{k+1}^\top,
\tag{6}
$$

*and $\mathbf{B}_0 = \mathbf{B}_{K+1} = \mathbf{0}$ since no cells exist below rank $0$ or above rank $K$. For CW complexes with uniform weights $\beta_k = 1$, these diagonal blocks reduce to the Hodge Laplacians: $\mathbf{D}_k = \Delta_k$.*

**Theorem 3.2** (PSD Condition). *The Graded Laplacian $\mathbf{L}^\star \succeq 0$ when*

$$
\gamma_k \leq \beta_{k+1} \cdot \sigma_{\min}^+(\mathbf{B}_{k+1}) \quad \text{for all } k = 0, \ldots, K-1,
\tag{7}
$$

*where $\sigma_{\min}^+(\cdot)$ denotes the smallest positive singular value.*

*Proof sketch.* We decompose the quadratic form $\mathbf{x}^\top \mathbf{L}^\star \mathbf{x} = \sum_{k=0}^{K-1} Q_k(\mathbf{x}_k, \mathbf{x}_{k+1})$, independent terms coupling adjacent ranks. Analyzing the condition $\mathbf{x}^T Q_k \mathbf{x} \geq 0$ through SVD yields $\gamma_k$ (see Appendix C.2). $\square$

Unlike Hodge Laplacians, which operate separately at each rank, $\mathbf{L}^\star$ is a single operator on the entire complex.

## 3.2. Rank Collapse by Schur Complement

While $\mathbf{L}^\star$ represents signals across all ranks, many downstream tasks only require vertex values. We therefore split a signal into the part we keep, $\mathbf{u} \in C^0$, and the higher-order part we eliminate, $\mathbf{z} \in \bigoplus_{k \geq 1} C^k$, and write

$$
\mathbf{L}^\star = \begin{pmatrix} \mathbf{A} & \mathbf{X} \\ \mathbf{X}^\top & \mathbf{C} \end{pmatrix}.
\tag{8}
$$

Here $\mathbf{A}$ is the vertex block, $\mathbf{C}$ contains the internal edge-, face-, and higher-cell dynamics, and $\mathbf{X}$ couples vertices to those cells. For a fixed vertex signal $\mathbf{u}$, the collapse chooses the higher-order values $\mathbf{z}$ that make the full energy as small as possible. If $\mathbf{C} \succ 0$, this relaxed higher-order completion is $\mathbf{z}^\star(\mathbf{u}) = -\mathbf{C}^{-1}\mathbf{X}^\top \mathbf{u}$, and substituting it back gives the vertex-only energy $\mathbf{u}^\top(\mathbf{A} - \mathbf{X}\mathbf{C}^{-1}\mathbf{X}^\top)\mathbf{u}$ via the Schur complement (Boyd & Vandenberghe, 2004). $\mathbf{X}\mathbf{C}^{-1}\mathbf{X}^\top$ records vertex-level energy that can be absorbed by

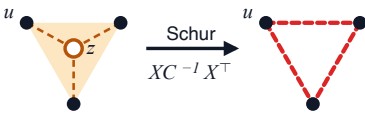

the eliminated higher-order cells, leaving a vertex operator that incorporates their structure. The singular case and regularized solve are addressed in sec. 3.4.

**Definition 3.3** (Collapsed Effective Operator). Given the block matrix in Eq. (8) with $\mathbf{C}$ invertible, where $\mathbf{A} \in \mathbb{R}^{n_0 \times n_0}$ operates on vertices and $\mathbf{C}$ on higher-order cells, we define the *collapsed effective operator* $\mathbf{S}$:

$$
\mathbf{S} := \mathbf{A} - \mathbf{X}\mathbf{C}^{-1}\mathbf{X}^\top
\tag{9}
$$

via the Schur complementation of $\mathbf{L}^\star$. This marginalizes higher-order structure to an operator at a single rank; from hereon, we assume $\mathbf{S}$ collapsed to the rank-0 (node) set due to practical relevance, although in theory any rank can be chosen by rearranging $\mathbf{L}^\star$ and defining $\mathbf{A}$ accordingly.

Let us consider a concrete construction of the collapsed operator from def. 3.3 for complexes with cells up to rank 2.

**Example 3.4** (Graded Laplacian for Rank-2 Complex). For a complex with maximum rank $K = 2$ (vertices, edges, faces), $\mathbf{L}^\star$ takes the explicit form:

$$
\mathbf{L}^\star = \begin{pmatrix}
\beta_1 \mathbf{B}_1 \mathbf{B}_1^\top & -\gamma_0 \mathbf{B}_1 & \mathbf{0} \\
-\gamma_0 \mathbf{B}_1^\top & \beta_1 \mathbf{B}_1^\top \mathbf{B}_1 + \beta_2 \mathbf{B}_2 \mathbf{B}_2^\top & -\gamma_1 \mathbf{B}_2 \\
\mathbf{0} & -\gamma_1 \mathbf{B}_2^\top & \beta_2 \mathbf{B}_2^\top \mathbf{B}_2
\end{pmatrix}
\tag{10}
$$

where $\beta_k$ are adjacency weights (diagonal blocks) and $\gamma_k$ are coupling weights (off-diagonal blocks). This can be partitioned as $\mathbf{L}^\star = \begin{pmatrix} \mathbf{A} & \mathbf{X} \\ \mathbf{X}^\top & \mathbf{C} \end{pmatrix}$ where:

- $\mathbf{A} = \beta_1 \mathbf{B}_1 \mathbf{B}_1^\top$ operates on vertices
- $\mathbf{C}$ governs higher-order cell dynamics
- $\mathbf{X} = \begin{pmatrix} -\gamma_0 \mathbf{B}_1 & \mathbf{0} \end{pmatrix}$ couples $\mathbf{A}$ to higher-order cells.

**Spectral Properties of the Collapsed Operator**. The Schur complement inherits fundamental spectral properties from the block Laplacian. We first establish the matrix inequalities, then interpret their physical consequences.

**Proposition 3.5** (Spectral Bounds). *Let $\mathbf{L}^\star \succeq 0$ with $\mathbf{C} \succ 0$. Then the collapsed effective operator satisfies:*

$$0 \preceq \mathbf{S} \preceq \mathbf{A}. \tag{11}$$

where $\mathbf{P} \preceq \mathbf{Q}$ means $\mathbf{Q} - \mathbf{P} \succeq 0$ is PSD (Boyd & Vandenberghe, 2004). This ordering has immediate consequences for eigenvalues. Let $0 = \lambda_1 \leq \lambda_2 \leq \cdots \leq \lambda_{n_0}$ denote the ordered eigenvalues of either operator.

**Corollary 3.6** (Eigenvalue Compression). *For each $k = 1, \ldots, n_0$:*

$$\lambda_k(\mathbf{S}) \leq \lambda_k(\mathbf{A}). \tag{12}$$

*Equality holds at index $k$ if and only if the corresponding eigenvector $\mathbf{v}_k$ of $\mathbf{A}$ satisfies $\mathbf{X}^\top \mathbf{v}_k = \mathbf{0}$.*

The gap is concentrated at the low end of the spectrum and closes at higher modes, where the eigenvectors $\mathbf{v}_k$ approach the null space of $\mathbf{X}^\top$.

The eigenvalue compression in Eq. (12) admits a physical interpretation. Recall that for a graph Laplacian $\mathbf{L}$, the effective conductance between nodes $i$ and $j$ can be characterized variationally (Doyle & Snell, 1984):

$$C_{ij}^{\mathbf{L}} = \min_{\mathbf{x}:x_i - x_j = 1} \mathbf{x}^\top \mathbf{L} \mathbf{x}. \tag{13}$$

**Corollary 3.7** (Reduction of Effective Conductance). *Higher-order cells reduce the effective conductance between vertices: $C_{ij}^{\mathbf{S}} \leq C_{ij}^{\mathbf{A}}$ for all $i, j \in V$.*

*Proof.* Since $\mathbf{S} \preceq \mathbf{A}$, any feasible $\mathbf{x}$ satisfies $\mathbf{x}^\top \mathbf{S} \mathbf{x} \leq \mathbf{x}^\top \mathbf{A} \mathbf{x}$. Taking minima preserves the inequality. $\square$

Physically, the inclusion of higher-order cells reduces the effective flow of information through the system. Vertices that share membership in high-rank cells experience weakened direct coupling due to the subtractive mediated term $\mathbf{X}\mathbf{C}^{-1}\mathbf{X}^\top$. This reflects a dynamic where higher-order routes interfere with or direct signal flow away from standard pairwise edges. When analyzing the properties of spectral operators, Cheeger inequalities also play an important role.

**Remark 3.8** (Cheeger Inequalities and Spectral Compression). The classical Cheeger inequality for graphs relates the spectral gap $\lambda_2(\mathbf{L}^G)$ to the Cheeger constant $h(G)$, measuring edge expansion (Alon & Milman, 1985):

$$\frac{h(G)^2}{2} \leq \lambda_2(\mathbf{L}^G) \leq 2h(G). \tag{14}$$

A small spectral gap implies weak connectivity and the presence of sparse cuts, while a strictly positive lower bound on $\lambda_2$ guarantees well-connected, fast-mixing dynamics. In the context of the collapsed effective operator $\mathcal{S}$, the spectral compression established in Proposition 3.5 yields:

$$\lambda_2(\mathbf{S}) \leq \lambda_2(\mathbf{A}) \leq 2h(G), \tag{15}$$

so the upper Cheeger bound transfers automatically to the effective operator $\mathbf{S}$. However, the lower bound does not transfer: since $\lambda_2(\mathbf{S}) \leq \lambda_2(\mathbf{A})$, the spectral gap may shrink under Schur reduction, potentially violating $\lambda_2(\mathbf{S}) \geq h(G)^2/2$. This mirrors a fundamental obstacle in extending Cheeger inequalities to higher-order structures Gundert & Szedlák (2014); Parzanchevski et al. (2016) (see Appendix C.1).

### 3.3. Realization on Combinatorial Complexes

So far, $\mathbf{B}_k$ has denoted a cellular boundary matrix. The same algebra also applies when these matrices are replaced by incidence matrices between ranks. However, in the case of more general structures such as Combinatorial Complexes (CCs), the homological interpretation changes.

CCs are a useful example because they allow higher-order cells without requiring all lower-rank subsets to be present (Hajij et al., 2022; 2023).

**Definition 3.9** (Combinatorial Complex and Incidence). CC is a triple $C = (S, \mathcal{X}, \mathrm{rk})$, where $S$ is a finite ground set, $\mathcal{X} \subseteq 2^S$ is a collection of non-empty subsets called cells, and $\mathrm{rk} : \mathcal{X} \rightarrow \mathbb{Z}_{\geq 0}$ is order-preserving: $x \subseteq y$ implies $\mathrm{rk}(x) \leq \mathrm{rk}(y)$. For $\mathcal{X}_k = \{\sigma \in \mathcal{X} : \mathrm{rk}(\sigma) = k\}$, we write $\mathbf{I}_{k,i}$ for the binary incidence matrix between ranks $k < i$:

$$[\mathbf{I}_{k,i}]_{pq} = 1 \quad \text{iff} \quad \sigma_p \subseteq \sigma_q, \tag{16}$$

where containment may be direct or transitive through intermediate cells.

If a CC contains incidences that skip ranks, one can pass to an adjacent-rank graded poset by inserting formal intermediate elements along those skipped relations. This "rank filling" is only a bookkeeping step for constructing adjacent-rank incidence matrices; we formalize it in Appendix A.3. With such an adjacent-rank realization, we instantiate Eq. (5) by replacing $\mathbf{B}_k$ with $\mathbf{I}_{k-1,k}$. The Schur complement and collapse are unchanged, but the diagonal blocks $\mathbf{D}_k$ become incidence-induced adjacency operators.

Next, we analyze practical considerations concerning the computability of the *collapsed effective operator* $\mathbf{S}$.

### 3.4. Scalable Computation and Regularization

A direct computation of $\mathbf{S}$ in Eq. (9) can be costly for large complexes. We address two practical considerations: (i) the matrix $C$ may be singular, and (ii) computing the inverse $\mathbf{C}^{-1}$ does not preserve sparsity. Of particular concern is the kernel of $\mathbf{C}$; scholars of topology may observe that $\ker(\Delta_k)$ contains the *homology cycles* of the rank with proper boundary operators (Hatcher, 2002). These cycles represent topological "holes" whose count at each dimension, the Betti

**Algorithm 1** Application of the Regularized Collapsed Higher-Order Operator

---

**Require:** Vertex signal $x$, boundary or incidence matrices $\{\mathbf{B}_k\}$, adjacency weights $\{\beta_k\}$, coupling weights $\{\gamma_k\}$, regularization $\epsilon$
**Ensure:** $\mathbf{S}_\epsilon x = (A - X(C + \epsilon I)^{-1} X^\top) x$
1: **Construct Blocks:**
2: $A \leftarrow \beta_1 \mathbf{B}_1 \mathbf{B}_1^\top, \quad X \leftarrow [-\gamma_0 \mathbf{B}_1, \mathbf{0}]$
3: $C \leftarrow$ Block-tridiagonal higher-order Laplacian
4: **Implicit Solve:**
5: $y \leftarrow X^\top x$
6: $z \leftarrow (C + \epsilon I)^{-1} y \qquad$ {Use sparse solver (e.g., CG)}
7: **return** $\mathbf{S}_\epsilon x \leftarrow Ax - Xz$

---

numbers, determines the kernel dimension of the Laplacian. We address this via Tikhonov regularization:

$$\mathbf{S}_\epsilon := \mathbf{A} - \mathbf{X}(\mathbf{C} + \epsilon \mathbf{I})^{-1} \mathbf{X}^\top. \tag{17}$$

Since $\mathbf{C} \succeq 0$, this only requires a small numerical correction; in practice we choose $\epsilon \in [10^{-6}, 10^{-4}]$, stabilizing the near-null part of the higher-order solve while leaving the positive spectrum close to the unregularized collapse. The following proposition makes the approximation precise.

**Proposition 3.10** (Regularized Collapse Error)**.** *Consider the rank-0 collapse of the Graded Laplacian in Eq. (5), with block decomposition $\mathbf{L}^\star = \begin{pmatrix} \mathbf{A} & \mathbf{X} \\ \mathbf{X}^\top & \mathbf{C} \end{pmatrix}$ and weights satisfying thm. 3.2. Let*

$$\mathbf{S}^\dagger := \mathbf{A} - \mathbf{X}\mathbf{C}^\dagger\mathbf{X}^\top, \tag{18}$$

*denote the zero-regularization limit, where $\mathbf{C}^\dagger$ is the Moore–Penrose pseudoinverse. Then the kernel of $\mathbf{C}$ does not contribute to the vertex-level operator. If $\mathbf{C}$ has positive eigenvalues, then for $\epsilon > 0$,*

$$\|\mathbf{S}_\epsilon - \mathbf{S}^\dagger\|_2 \le \frac{\epsilon \|\mathbf{X}\|_2^2}{\lambda_+(\lambda_+ + \epsilon)}. \tag{19}$$

*where $\lambda_+ := \lambda_{\min}^+(\mathbf{C})$ is the smallest positive eigenvalue of $\mathbf{C}$. If $\mathbf{C}$ has no positive eigenvalues, then instead $\mathbf{X} = \mathbf{0}$ and $\mathbf{S}_\epsilon = \mathbf{S}^\dagger = \mathbf{A}$. In particular, $\mathbf{S}_\epsilon \to \mathbf{S}^\dagger$ as $\epsilon \to 0$; if $\mathbf{C}$ is nonsingular, this limit is the exact Schur complement $\mathbf{S}$.*

The proof is provided in Appendix C.3. Thus, the regularization does not introduce an uncontrolled $1/\epsilon$ contribution from homological kernel directions; it only damps the positive higher-order modes, and the induced spectral perturbation is bounded by Eq. (19). By Weyl's inequality, the same quantity bounds the change in each eigenvalue. We use the regularized operator $\mathbf{S}_\epsilon$ in practice rather than forming $\mathbf{C}^\dagger$: the pseudoinverse is generally dense and sensitive to numerical rank decisions, whereas $\mathbf{C} + \epsilon \mathbf{I}$ is invertible and can be applied through sparse linear solves. To avoid densifying the entire operator $\mathbf{S}_\varepsilon$, we compute its action $\mathbf{S}_\varepsilon \mathbf{x}$

on a signal $\mathbf{x}$ iteratively. This procedure is summarized in Algorithm 1: note that because $\mathbf{C}$ retains the sparsity structure inherited from the boundary/incidence matrices, this step can be efficiently handled, e.g., via (preconditioned) Conjugate Gradient (CG).

# 4. Empirical Analysis

We evaluate the proposed *collapsed effective operator* by analyzing how higher-order structure, once collapsed to the vertex level, affects empirical quantities such as spectral properties, diffusion behavior, and ability to characterize local neighborhoods. Rather than treating the operator $\mathbf{S}_\varepsilon$ as a replacement for existing Laplacians, our experiments are designed to probe its quantitative effects in several ways. We begin with controlled synthetic complexes to study signal smoothness and energy behavior (sec. 4.1), examine spectral clustering and segmentation tasks (secs. 4.2 and 4.3), and finally assess downstream learning performance using spectral features as input (secs. 4.4 and 4.5).

We compare to the Multiorder Laplacian (Lucas et al., 2020) as a natural alternative: it builds one vertex Laplacian per hyperedge order and sums these contributions, whereas our operator first couples ranks in a graded block operator and then collapses them by Schur complement.

**Definition 4.1** (Multiorder Laplacian (Lucas et al., 2020))**.** Let $\mathcal{H}$ be a hypergraph on $n_0$ vertices with maximum hyperedge order $d_{\max}$. For each order $d \ge 1$, define the *d-th order Laplacian* $\mathbf{L}^{(d)} \in \mathbb{R}^{n_0 \times n_0}$ on vertices by

$$L_{ij}^{(d)} = d K_i^{(d)} \delta_{ij} - A_{ij}^{(d)}, \tag{20}$$

where $K_i^{(d)}$ is the generalized degree of vertex $i$ at order $d$ and $A_{ij}^{(d)}$ counts order-$d$ hyperedges containing both $i$ and $j$. The *Multiorder Laplacian* $\mathbf{L}^{(\mathrm{mult})}$ aggregates contributions across orders:

$$L_{ij}^{(\mathrm{mult})} = \sum_{d=1}^{d_{\max}} \frac{\gamma_d}{\langle K^{(d)} \rangle} L_{ij}^{(d)}, \tag{21}$$

where $\gamma_d > 0$ weights each order and $\langle K^{(d)} \rangle$ denotes the average $d$-th order generalized degree.

For normalizing cuts, we performed grid search over $\beta_1 \in \{0.1, 1, 10\}$ and $\varepsilon \in \{0.1, 0.5, 1\}$ for $\mathbf{S}_\varepsilon$, and $\gamma_0, \gamma_1 \in \{0.1, 0.5, 1.0, 2.0, 5.0\}$ for $\mathbf{L}^{(\mathrm{mult})}$. The optimal parameters are reported in Appendix B.1. For all other experiments, we fix $\beta_0 = \beta_1 = \beta_2 = 1$ with $\gamma_0, \gamma_1$ set to the upper bound from thm. 3.2 to hold, and all $\gamma$'s to 1 for $\mathbf{L}^{(\mathrm{mult})}$.

## 4.1. Signal Smoothness

We first assess the collapsed effective operator $\mathbf{S}_\varepsilon$ as a **topologically aware filter**, comparing it against the standard

**Table 1.** Manifold denoising on random geometric graphs with increasing topological noise. We report reconstruction MSE under 0, 100, 150, and 200 randomly added shortcut edges. The collapsed operator $S_\epsilon$ remains robust as graph locality is degraded.

| Exp. | Metric | $\mathbf{L^G}$ | $\mathbf{L}^{(\text{mult})}$ | $\mathbf{S}_\varepsilon$ |
|---|---|---|---|---|
| Manifold (0) | MSE ↓ | **0.066** | 0.074 | **0.066** |
| Manifold (100) | MSE ↓ | 0.082 | 0.081 | **0.074** |
| Manifold (150) | MSE ↓ | 0.092 | 0.088 | **0.072** |
| Manifold (200) | MSE ↓ | 0.099 | 0.094 | **0.073** |

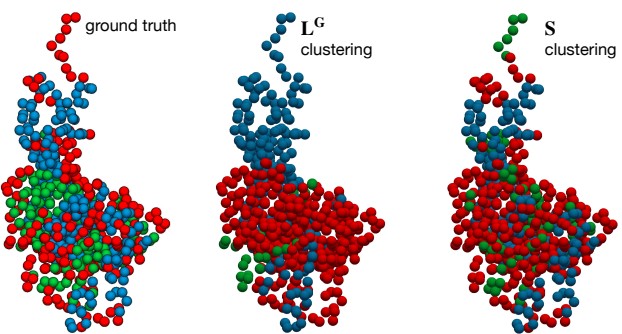

**Figure 3.** Unsupervised protein segmentation on 1LYZ from the Topotein benchmark (Wang et al., 2025). Left: Ground truth. Middle: The standard Graph Laplacian (46.9%) fails to capture interleaved motifs. Right: Our effective operator (70.9%) successfully recovers these non-local structures, significantly outperforming the baseline.

graph Laplacian $\mathbf{L^G}$ and the Multiorder Laplacian $\mathbf{L}^{(\text{mult})}$ (def. 4.1, def. 3.3) on a manifold denoising task: signal reconstruction on random geometric graphs ($N = 200$) with Gaussian feature noise (Table 1). Denoising a signal on a proximity graph is the discrete analogue of denoising on a manifold, with image denoising, where, e.g., pixels form nodes of a similarity graph (Elmoataz et al., 2008; Pang & Cheung, 2017). We inject controlled topological noise by adding 0, 100, 150, 200 random shortcut edges between arbitrary node pairs, progressively breaking the graph's geometric locality, and measure each operator's sensitivity to spurious non-local connectivity. While all operators are tied in the clean regime ($\mathbf{S}_\varepsilon$ and $\mathbf{L^G}$ both attain 0.066), their behavior diverges as locality degrades: $\mathbf{L^G}$ deteriorates by roughly 50% (0.066 → 0.099), whereas $\mathbf{S}_\varepsilon$ remains essentially flat (0.066 → 0.073). This robustness follows directly from thm. 3.7: shortcut edges that close into higher-order cells have their pairwise coupling damped by $\mathbf{X}(\mathbf{C} + \epsilon\mathbf{I})^{-1}\mathbf{X}^\top$, so $\mathbf{S}_\varepsilon$ selectively suppresses exactly the spurious couplings that $\mathbf{L^G}$ over-smooths across.

### 4.2. Spectral HKS Clustering on CC

Spectral clustering partitions graphs using the eigenvectors of the Laplacian, making it a natural testbed for evaluating whether our collapsed effective operator captures topologically meaningful structure that improves cluster separability relative to standard graph-based methods.

**Setup**. We evaluate our method on the Topotein benchmark (Wang et al., 2025) by comparing to the standard graph Laplacian $\mathbf{L^G}$. For each protein, we construct a combinatorial complex composed of Rank-0 cells (individual residues) and Rank-1 cells (backbone connections and hydrogen bond edges with N-O distance < 3.5 Å). The complex is capped with Rank-2 cells which are derived from local cliques.

The ground truth labels of the complex are derived from Secondary Structure Elements (SSEs), adopting the standard three-state classification defined by DSSP (Kabsch & Sander, 1983): $\alpha$-helices (H), characterized by stabilized spiral backbone conformations; $\beta$-sheets (E), consisting of extended parallel or anti-parallel strands; and Coils (C), representing flexible loops and unstructured regions.

Since our ground-truth clusters are distributed throughout the shape, as observed in fig. 3 (left), we perform the clustering on the HKS features induced by the corresponding operator. This yields a multi-scale feature vector per residue that is then clustered via k-means ($k = 3$: Helix, Sheet, Coil). We report accuracy via Hungarian matching.

**Results**. Our method achieves 70.9% accuracy versus 46.9% for the baseline: a 24-point improvement. Further fig. 3 visualizes a representative protein: the Graph Laplacian clusters by spatial proximity (middle), creating large uniform regions, while our effective operator $\mathbf{S}_\varepsilon$ (right) correctly recovers the fragmented, interleaved SSE structure (left). The complement's induced connectivity within Rank-2 cells enables spectral separation of topologically distinct SSEs even when spatially adjacent.

### 4.3. Normalizing Cuts on Lifted Graphs

**Setup**. We evaluate the effective operator on graph clustering using normalized cuts on two benchmarks: synthetic stochastic block models (SBM) (Holland et al., 1983) and the *real-world* networks: Karate (Zachary, 1977), Football (Girvan & Newman, 2002), Misérables (Knuth & Knuth, 1993), Books (Krebs, 2004) and Dolphins (Lusseau et al., 2003).

Each graph is represented as a combinatorial complex with Rank-0 cells (nodes), Rank-1 cells (edges), and Rank-2 cells (triangles and tetrahedra when present). $\mathbf{S}_\varepsilon$ and $\mathbf{L}^{(\text{mult})}$ incorporate higher-order connectivity, while the baseline uses the standard graph Laplacian $\mathbf{L^G}$. We apply spectral clustering with normalized cuts and report clustering accuracy.

**Results**. Table 2 and Table 3 show consistent improvements of the collapsed operator over the standard Laplacian $\mathbf{L^G}$. On synthetic SBM benchmarks (Table 2), our method achieves substantial gains on Dense-Cliques +3% and Unequal-Sizes + 6%. $\mathbf{S}_\varepsilon$ particularly excels when

**Table 2.** SBM (Holland et al., 1983) results. Best in **bold**. Base=Baseline, Den=Dense-Cliques, +N=Cliques+Noise, Plt=Planted-Clique, SD=Small-Dense, LS=Large-Sparse, Uneq=Unequal-Sizes, Br=Bridge-Only.

| | Base | Den | +N | Plt | SD | LS | Uneq | Br |
|---|---|---|---|---|---|---|---|---|
| $N$ | 80 | 80 | 80 | 80 | 72 | 120 | 90 | 80 |
| $E$ | 363 | 554 | 630 | 385 | 340 | 400 | 597 | 452 |
| $T_3$ | 174 | 1580 | 1053 | 941 | 594 | 103 | 899 | 940 |
| $T_4$ | 21 | 2261 | 789 | 2667 | 572 | 4 | 556 | 859 |
| $\mathbf{L^G}$ | .58 | .96 | .70 | **.96** | **.74** | .77 | .92 | **1.0** |
| $\mathbf{L^{(mult)}}$ | .61 | .90 | .73 | .65 | .70 | **.82** | .91 | .98 |
| $\mathbf{S_\varepsilon}$ | **.61** | **.98** | **.84** | **.96** | **.74** | .78 | **.98** | **1.0** |

**Table 3.** Performance on *real-world* datasets (best in **bold**). Clustering accuracy is reported.

| Metric | Karate | Football | Misérables | Books | Dolphins |
|---|---|---|---|---|---|
| $N$ | 34 | 115 | 77 | 105 | 62 |
| $E$ | 78 | 613 | 254 | 441 | 159 |
| $T_3$ | 45 | 810 | 467 | 560 | 95 |
| $T_4$ | 11 | 732 | 639 | 319 | 27 |
| $\mathbf{L^G}$ | **0.97** | 0.47 | 0.64 | **0.82** | **0.69** |
| $\mathbf{L^{(mult)}}$ | 0.94 | 0.46 | 0.52 | 0.73 | **0.69** |
| $\mathbf{S_\varepsilon}$ | **0.97** | **0.50** | **0.66** | **0.82** | **0.69** |

higher-order structures (triangles $T_3$, tetrahedra $T_4$) are abundant, as these Rank-2 cells induce Schur-mediated damping of within-cell pairwise couplings, reshaping the low-frequency spectrum and improving cluster separability. On real-world networks (Table 3), we observe moderate but consistent improvements: +3% on College Football, +2% on Les Misérables. $\mathbf{S_\varepsilon}$ effectively leverages higher-order topology to guide spectral partitioning.

## 4.4. Spectrum Regression

**Setup.** We evaluate spectral features on the Mantra dataset (Ballester et al., 2025), a benchmark for distinguishing different simplicial complexes. We compare three spectral feature extraction methods: **spectrum $\mathbf{L^G}$** (eigenvalues of the standard graph Laplacian), **spectrum $\mathbf{L^{(mult)}}$** (eigenvalues of the multi Laplacian), and **spectrum $\mathbf{S_\varepsilon}$** (eigenvalues of the collapsed effective operator). An MLP then classifies the Betti numbers of each complex. We report mean accuracy ± standard deviation over five runs.

**Results.** Table 5 shows that all three spectral methods achieve perfect classification on the simple $\beta_0$ class (1.00 accuracy). However, the spectrum of $\mathbf{S_\varepsilon}$ significantly outperforms the standard graph spectrum on the more challenging $\beta_1$ class: 0.95 vs. 0.92 accuracy for the graph Laplacian, and 0.97 for multi-laplacian. On $\beta_2$, the collapsed operator maintains competitive performance (0.95) while the graph Laplacian achieves 0.92, and the multi-laplacian reaches 0.97. Notably, all topological methods substantially outperform standard graph neural networks (GAT and GCN

achieve only 0.31-0.33 on $\beta_1$), demonstrating that spectral signatures are sufficient to classify these quantities, results that, to the best of our knowledge, have not been reported.

**Discussion on Operator Trade-offs.** The collapsed operator is specifically designed for regimes where higher-order topological structure carries a discriminative signal for vertex-level tasks, such as tertiary structure in proteins Appendix B.3 and normalized cuts sec. 4.1. In these regimes, the collapsed operator outperforms the graph Laplacian by explicitly aggregating cross-rank interactions into a vertex-level representation. The inherent trade-off is that collapsing across ranks does not fully preserve rank-local topological invariants, such as individual Betti numbers. On benchmarks like MANTRA, where tasks depend heavily on rank-local signals, rank-wise methods that retain the full Hodge structure (e.g., the multi-order Laplacian) are naturally better suited. Nevertheless, our results demonstrate that the collapsed operator retains substantially more topological information than the standard graph Laplacian, striking a practical balance for node-level applications.

### 4.5. Node Classification on Proteins

We evaluate our operator on the vertex-level task of *node classification* on proteins, a setting where non-local topological structure provides a useful inductive bias (Table 4).

**Experimental Setup.** We construct a CC for proteins (sec. 4.2) and perform transductive node classification (analogous to GCN on Cora (Kipf & Welling, 2017)). We classify amino acids based on their relative molecular position using two label sets: *Contact* (3 structural classes) and *ResType* (23 classes based on amino acid type). We compare positional encodings (PE) derived from the standard graph Laplacian against those from our operator, $\mathbf{S_\varepsilon}$ (**SchurPE**), integrated into GCN (Kipf & Welling, 2017) and Graph-Transformer (GT) (Dwivedi & Bresson, 2021).

**Table 4.** Transductive node classification accuracy on protein molecular structures. Best in **bold**.

| Model + PE | Contact | ResType |
|---|---|---|
| GCN + NoPE | 0.471 ± 0.013 | 0.524 ± 0.019 |
| GT + NoPE | 0.463 ± 0.034 | 0.773 ± 0.019 |
| GCN + LaplacePE | 0.480 ± 0.010 | 0.582 ± 0.033 |
| GT + LaplacePE | 0.551 ± 0.004 | 0.762 ± 0.028 |
| **GCN + SchurPE** | **0.494 ± 0.013** | **0.666 ± 0.081** |
| **GT + SchurPE** | **0.573 ± 0.009** | **0.789 ± 0.019** |

**Results.** As shown in Table 4, SchurPE consistently improves performance across both architectures and classification tasks. The significant gains indicate that $\mathbf{S_\varepsilon}$ provides superior signals for capturing structural nuances compared to standard Laplacian embeddings. This suggests that our

**Table 5.** Results on determining the betti number of the complexes defined in the Mantra dataset ($2 - \mathcal{M}^0$) (Ballester et al., 2025).

| | Accuracy | | |
|---|---|---|---|
| MODEL (CLASS) | $\beta_0$ | $\beta_1$ | $\beta_2$ |
| MLP | **1.00 ± 0.00** | 0.31 ± 0.00 | 0.92 ± 0.00 |
| GAT (Veličković et al., 2018) | **1.00 ± 0.00** | 0.31 ± 0.00 | 0.92 ± 0.00 |
| GCN (Kipf & Welling, 2017) | **1.00 ± 0.00** | 0.31 ± 0.00 | 0.92 ± 0.00 |
| TAG (Du et al., 2017) | **1.00 ± 0.00** | 0.32 ± 0.01 | 0.92 ± 0.00 |
| UniMP (Shi et al., 2021) | **1.00 ± 0.00** | 0.33 ± 0.00 | 0.92 ± 0.00 |
| CellMP (Bodnar et al., 2021a) | 0.46 ± 0.50 | 0.39 ± 0.35 | 0.46 ± 0.44 |
| CT (Ballester et al., 2024) | **1.00 ± 0.00** | 0.93 ± 0.00 | 0.93 ± 0.00 |
| DECT (Roell & Rieck, 2024) | **1.00 ± 0.00** | 0.32 ± 0.00 | 0.92 ± 0.00 |
| SAN (Goh et al., 2022) | 0.09 ± 0.04 | 0.12 ± 0.10 | 0.52 ± 0.14 |
| SCCN (Yang et al., 2022) | **1.00 ± 0.00** | 0.93 ± 0.00 | 0.93 ± 0.00 |
| SCCNN (Yang & Isufi, 2023) | 0.00 ± 0.00 | 0.03 ± 0.02 | 0.33 ± 0.37 |
| SCN (Wu et al., 2023) | 0.33 ± 0.38 | 0.21 ± 0.26 | 0.62 ± 0.36 |
| Spectrum $\mathbf{L^G}$ | **1.00 ± 0.00** | 0.92 ± 0.00 | 0.92 ± 0.00 |
| Spectrum $\mathbf{L^{(mult)}}$ | **1.00 ± 0.00** | **0.97 ± 0.00** | **0.97 ± 0.00** |
| Spectrum $\mathbf{S_\varepsilon}$ | **1.00 ± 0.00** | 0.95 ± 0.00 | 0.95 ± 0.00 |

proposed operator captures useful higher-order information beyond simple clustering, which is particularly salient for node-level prediction tasks.

**Computational Complexity**. Our computational improvements stem from architectural bottlenecks of HOMP (Hajij et al., 2022). Let $M$ denote total incidences across all ranks, $m$ the edge count, and $d$ the feature dimension. A single HOMP layer costs $O(Md^2)$ as it propagates features across all cells, whereas a GNN layer costs $O(md^2)$. Our approach amortizes this cost to pre-processing: each forward pass thus achieves $O(M/m)$ speedup, since $M \gg m$.

## 5. Related Work

**Higher-Order Laplacians**. The Hodge Laplacian (Eckmann, 1944) generalises spectral graph theory to simplicial complexes through boundary operators, forming the basis for signal processing on $k$-chains (Barbarossa & Sardellitti, 2020; Schaub et al., 2021). Recent works (Huang et al., 2024; Mémoli et al., 2022) have generalized the Hodge Laplacian to overcome the representational shortcomings of the standard definition. The Multiorder Laplacian (Lucas et al., 2020) sums simplex contributions but models them as additive stiffness rather than mediating conductance. In (Calmon et al., 2023) the Laplacian spaces are coupled together. However, the resulting operator is not necessarily PSD nor does it reduce to the vertex space. Our proposed *effective operators* depart from these approaches by marginalizing higher-order ranks into a single operator.

**Schur Complements on Graphs**. The Schur complement has found analogous applications in electrical networks (i.e., effective resistance) (Babić et al., 2002; Devriendt, 2022; Dorfler & Bullo, 2012) and Gaussian graphical models through marginalization (Lauritzen, 1996). We repurpose this tool for a different goal: collapsing *topological ranks* rather than spatial subsets, producing an implicit vertex

operator that captures higher-order interactions.

**Topological Deep Learning**. Topological Deep Learning lifts representation learning from graphs to simplicial, cellular, and combinatorial complexes. Higher-order message passing (HOMP) was introduced for simplicial complexes (Bodnar et al., 2021b) and later generalized to broader complex classes (Bodnar et al., 2021a; Hajij et al., 2022), improving the expressivity of message-passing neural networks beyond pairwise interactions. Recent work has focused on unifying benchmarks and architectures for higher-order learning (Telyatnikov et al., 2025; Papillon et al., 2025).

**Spectral Positional Encodings**. Positional encodings enhance GNN expressivity by providing nodes with structural information beyond local neighborhoods. Spectral approaches based on Laplacian eigenfunctions are particularly effective: Laplacian eigenvectors serve as positional encodings (Dwivedi & Bresson, 2021; Kreuzer et al., 2021) and have been extended to higher-order spaces (Barsbey et al., 2025), while eigenvalue-weighted combinations yield multi-scale geometric descriptors. Heat Kernel Signatures (HKS) (Sun et al., 2009) and its graph application (Donnat et al., 2018) are encoding geometry through heat diffusion at multiple time scales. More broadly, Laplacian-based kernels such as RBF and diffusion kernels are widely used in graph signal processing (Smola & Kondor, 2003).

## 6. Concluding Remarks

We introduce *Collapsed Effective Operators* as a framework to translate higher-order topological structures into effective vertex-level dynamics. Unlike additive approaches that treat each rank independently, our method captures the emergent couplings induced in higher-order cells at the vertex level. Our theoretical analysis revealed that the resulting operators encode non-local topological interactions and have an interpretable spectral and variational structure. Empirically, we demonstrate this collapsed operator is useful to analyze node-level dynamics in various settings.

**Limitations**. The Schur complement does not preserve the full graded spectrum: rank-local invariants may be partially lost, and the spectral gap can shrink under reduction (rem. 3.8), leaving Cheeger-type lower bounds for **S** open. Moreover, **S** is typically dense, increasing explicit construction costs despite admitting implicit application (sec. 3.4). Finally, we use scalar per-rank weights $\beta_k, \gamma_k$ rather than metric-aware per-cell weights; incorporating Hodge-star, cotangent, or distance-based weights is a natural extension. Further directions include time-varying topologies (Einizade et al., 2025) and end-to-end learned effective operators.

**Acknowledgements**. The authors would like to thank the ICML reviewers and Mustafa Hajij for their helpful feedback on preliminary versions of the manuscript. The authors are also grateful for support from the UK AI Research Resource (AIRR) through grant 0251-4584-0945-1. T. B. was supported by the UKRI Engineering and Physical Sciences Research Council (EPSRC) through the Future Leaders Fellowship [grant number MR/Y018818/1]. L.B. was supported by the UK Royal Society through grant NIF/R1/254128. M.K. and V.G. acknowledge support from the Finnish Doctoral Program Network in Artificial Intelligence (AI-DOC).

## Impact Statement

This work introduces a spectral operator for higher-order structures. Although the contribution is primarily theoretical, we demonstrate its practical utility in spectral clustering, protein structure segmentation, and node-level molecular and protein representation learning.

Potential risks are those shared broadly by methodological advances in topological deep learning: improved structural methods could, in principle, facilitate privacy violations in social network analysis or adversarial applications in drug design. As an indirect, methodological contribution, we view these concerns as community-wide rather than specific to our work, and we advocate for responsible deployment.

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

## Table of Contents

## A. Formal Definitions: Simplicial, CW and Combinatorial Complexes

Combinatorial complexes are introduced as generalized topological data structures in (Hajij et al., 2023; 2022) and refer the reader to (Boissonnat et al., 2018; Chazal & Michel, 2021; Zomorodian, 2012; Hatcher, 2002) for comprehensive treatments of algebraic topology We provide a brief review below.

**Definition A.1** (Simplicial Complex). A simplicial complex $\mathcal{S}$ a complex where every cell is a simplex. Formally, $\mathcal{S}$ is a finite collection of sets (simplices) closed under the subset operation: if $\sigma \in \mathcal{S}$ and $\tau \subseteq \sigma$, then $\tau \in \mathcal{S}$. A $k$-simplex is a set of cardinality $k + 1$.

The boundary operator is given explicitly:

$$\partial_k([v_0, \ldots, v_k]) = \sum_{i=0}^{k} (-1)^i [v_0, \ldots, \hat{v}_i, \ldots, v_k], \tag{22}$$

where $\hat{v}_i$ denotes omission of vertex $v_i$. This ensures $\partial_{k-1}\partial_k = 0$ algebraically.

**Definition A.2** (CW Complex). A CW complex $\mathcal{K}$ is a topological space constructed inductively by attaching cells. Formally, $\mathcal{K}$ is equipped with:

- A filtration $\mathcal{K}^0 \subseteq \mathcal{K}^1 \subseteq \cdots \subseteq \mathcal{K}^d = \mathcal{K}$, where $\mathcal{K}^k$ is the $k$-skeleton.

- For each $k$-cell $e_\alpha^k$, a characteristic map $\Phi_\alpha : D^k \to \mathcal{K}^k$ from the closed $k$-disc $D^k$ such that:
  1. $\Phi_\alpha$ restricts to a homeomorphism $\text{int}(D^k) \to e_\alpha^k$.
  2. The attaching map $\Phi_\alpha|_{\partial D^k} : S^{k-1} \to \mathcal{K}^{k-1}$ maps the boundary into the $(k-1)$-skeleton.

**Definition A.3** (Combinatorial Complex). A Combinatorial Complex (CC) $C$ is a ranked set-system $C = (X, \{X^k\}_{k=0}^d, \subseteq)$ where:

- $X$ is a finite ground set (typically vertices).

- $X^k \subseteq 2^X$ is the collection of **rank-$k$ cells** (subsets of $X$).

- The **rank function** rank $: 2^X \to \mathbb{N}$ assigns each cell its dimension.

- **No closure requirement**: if $\sigma \in X^k$, subsets $\tau \subset \sigma$ need not belong to $C$.

The **unsigned incidence matrix $\mathbf{B_k} \in \{0, 1\}^{|X^{k-1}| \times |X^k|}$** encodes containment:

$$(\mathbf{B_k})_{ij} = \begin{cases} 1 & \text{if cell } \sigma_i^{k-1} \subseteq \sigma_j^k, \\ 0 & \text{otherwise.} \end{cases} \tag{23}$$

In general, $\mathbf{B_{k-1}B_k} \neq \mathbf{0}$, as composition $\mathbf{B}_{k-1}\mathbf{B_k}$ counts paths of membership through intermediate ranks, not boundaries. When restricted to simplices, CCs recover simplicial complexes with $\mathbf{B}_k = |\mathbf{D}_k|$ (absolute values).

**Remark A.4.** The distinction is fundamental: CW/simplicial complexes satisfy $\partial_{k-1}\partial_k = 0$ (topological closure), enabling Hodge theory. Combinatorial complexes permit $\mathbf{B}_{k-1}\mathbf{B}_k \neq 0$ (hierarchical co-occurrence), which is suitable for modelling non-conservative processes such as information diffusion or chemical reaction networks.

## A.1. The Schur Complement

A key tool in our approach to constructing an effective higher-order operator on the vertex set is the Schur complement, a fundamental linear-algebraic concept with important applications in convex optimization. We follow the notation of (Boyd & Vandenberghe, 2004), and refer the reader to this foundational text for a thorough treatment.

**Definition A.5** (Schur Complement). Let $\mathbf{M} \in \mathbb{R}^{(n+m)\times(n+m)}$ be a block matrix of the form

$$\mathbf{M} = \begin{bmatrix} \mathbf{A} & \mathbf{X} \\ \mathbf{Y} & \mathbf{D} \end{bmatrix}, \tag{24}$$

where $\mathbf{A} \in \mathbb{R}^{n\times n}$, $\mathbf{X} \in \mathbb{R}^{n\times m}$, $\mathbf{Y} \in \mathbb{R}^{m\times n}$, and $\mathbf{D} \in \mathbb{R}^{m\times m}$. If $\mathbf{D}$ is invertible, the *Schur complement* of $\mathbf{D}$ in $\mathbf{M}$ is defined as

$$\mathbf{M}/\mathbf{D} \triangleq \mathbf{A} - \mathbf{X}\mathbf{D}^{-1}\mathbf{Y}. \tag{25}$$

A key result connects the positive definiteness of block matrices to their Schur complements.

**Theorem A.6** (Schur Complement Lemma). *Let $\mathbf{M}$ be a symmetric block matrix*

$$\mathbf{M} = \begin{bmatrix} \mathbf{A} & \mathbf{X} \\ \mathbf{X}^\top & \mathbf{C} \end{bmatrix}, \tag{26}$$

*with $\mathbf{A} \in \mathbb{S}^n$ and $\mathbf{C} \in \mathbb{S}^m$. Then the following equivalences hold:*

1. $\mathbf{M} \succ 0$ *if and only if* $\mathbf{A} \succ 0$ *and* $\mathbf{C} - \mathbf{X}^\top\mathbf{A}^{-1}\mathbf{X} \succ 0$.

2. $\mathbf{M} \succ 0$ *if and only if* $\mathbf{C} \succ 0$ *and* $\mathbf{A} - \mathbf{X}\mathbf{C}^{-1}\mathbf{X}^\top \succ 0$.

*For positive semi-definiteness, if $\mathbf{A} \succeq 0$, then*

$$\mathbf{M} \succeq 0 \iff \mathbf{A} \succeq 0, \ \mathbf{C} - \mathbf{X}^\top\mathbf{A}^\dagger\mathbf{X} \succeq 0, \ (\mathbf{I} - \mathbf{A}\mathbf{A}^\dagger)\mathbf{X} = 0, \tag{27}$$

*where $\mathbf{A}^\dagger$ denotes the Moore–Penrose pseudoinverse.*

## A.2. Incidence Structures of the Graded Laplacian: Homological vs. Combinatorial

The structural behavior of the proposed Graded Laplacian depends critically on the choice of rank-to-rank operator $\mathbf{B}_k$ mapping signals from rank $k$ to rank $k-1$. In the main framework, $\mathbf{B}_k$ is a cellular boundary matrix on a CW complex. For the incidence-based realization on combinatorial complexes, the same block algebra can instead use unsigned incidence matrices. This distinction separates two different topological priors:

**The Homological Setting (Simplicial/CW Complexes).** This setting recovers standard algebraic topology, where $\mathbf{B}_k$ is defined as the **signed boundary matrix $\mathbf{D}_k$**.

- **Chain Property:** The structure satisfies the fundamental algebraic constraint $\mathbf{D}_{k-1}\mathbf{D}_k = 0$, reflecting the geometric principle that the boundary of a boundary is empty.

- **Hodge Theory:** This orthogonality enables the Hodge decomposition of the resulting Laplacian, making this setting ideal for modelling conservative flows, fluxes, and homological features (holes, voids).

**The Combinatorial Setting (e.g., including set relationship).** This setting relaxes geometric constraints to model general set-based interactions. Here, $\mathbf{B}_k$ is defined as the **unsigned incidence matrix $\mathbf{I}_k$**, where $(I_k)_{ij} = 1$ if cell $j$ contains sub-cell $i$.

**Figure 4.** Rank completion for a skipped containment relation in a CC. A direct relation $x \prec y$ with a rank gap is replaced by a chain through formal intermediate elements, producing adjacent-rank cover relations for the incidence matrices used by the Graded Laplacian.

- **Generalized Connectivity:** Unlike the homological case, this setting allows for $\mathbf{I}_{k-1}\mathbf{I}_k \neq \mathbf{0}$. This non-vanishing product captures hierarchical co-occurrences (e.g., a node belonging to an edge that belongs to a face) that drive synchronisation and contagious processes rather than conservative flows.

- **No Closure:** The complex requires no geometric closure, allowing for flexible modelling of arbitrary higher-order relationships.

### A.3. Rank Completion of Combinatorial Complexes

In the main text we describe a "rank filling" step to apply adjacent-rank incidence operators to CCs with skipped ranks. Fig. 4 illustrates this operation for a relation that skips two ranks.

Let $C = (S, \mathcal{X}, \mathrm{rk})$ be a CC and let $\mathcal{R} \subseteq \mathcal{X} \times \mathcal{X}$ be the directed containment relations we choose to encode, whose transitive closure gives the containment relation of interest. If $(x, y) \in \mathcal{R}$ with $a = \mathrm{rk}(x)$, $b = \mathrm{rk}(y)$, and $b > a + 1$, replace this skipped relation by a chain

$$x = z_a \prec z_{a+1}^{(x,y)} \prec \cdots \prec z_{b-1}^{(x,y)} \prec z_b = y, \tag{28}$$

where each $z_\ell^{(x,y)}$ is a formal element of rank $\ell$. Adjacent-rank relations are kept unchanged. Applying this replacement to all skipped relations gives a finite graded poset $\widehat{C}$ with rank sets $\widehat{\mathcal{X}}_k$ and cover relations only between adjacent ranks.

The adjacent incidence matrices used in Eq. (5) are then

$$[\widehat{\mathbf{I}}_{k-1,k}]_{pq} = 1 \iff \hat{\sigma}_p \prec \hat{\tau}_q, \tag{29}$$

for $\hat{\sigma}_p \in \widehat{\mathcal{X}}_{k-1}$ and $\hat{\tau}_q \in \widehat{\mathcal{X}}_k$. This construction preserves reachability between original cells: $x$ is contained in $y$ in the encoded relation if and only if there is a directed chain from $x$ to $y$ in $\widehat{C}$. It does not add a boundary map or a homological interpretation; it only supplies adjacent-rank incidence matrices. Consequently, replacing $\mathbf{B}_k$ by $\widehat{\mathbf{I}}_{k-1,k}$ yields incidence-induced adjacency blocks, not Hodge Laplacians.

## B. Additional Information on experiments

### B.1. Hyperparameters in Normalizing Cuts

In the following section, we report the selected parameters for each experimental setup. Table 6 contains the optimal hyperparameters on the real-world dataset and Table 6b contain the optimal hyperparameters on the SBM datasets.

### B.2. Spectral HKS Clustering

To capture multi-scale geometric properties in the clustering of graph-structured domains, we employ Heat Kernel Signatures (HKS) (Sun et al., 2009), a useful means for extending spectral positional encodings inspired by heat diffusion.

**Mathematical Foundation**. Consider the heat equation on a graph with discrete Laplacian $\mathbf{L}$: $\frac{\partial u}{\partial t} = -\mathbf{L}u$.

**Table 6.** Optimal hyperparameters selected via grid search to maximise accuracy.

**(a)** Real-world datasets

| Dataset | Multi-Order | | Collapsed Eff. | |
|---|---|---|---|---|
| | $\gamma_0$ | $\gamma_1$ | $\varepsilon$ | $\beta_1$ |
| Karate Club | 0.1 | 0.1 | 0.5 | 1.0 |
| Football | 5.0 | 0.1 | 0.1 | 1.0 |
| Les Miserables | 0.1 | 0.1 | 1.0 | 10.0 |
| Political Books | 0.5 | 0.1 | 0.1 | 0.1 |
| Dolphins | 0.1 | 0.1 | 0.5 | 10.0 |

**(b)** Stochastic Block Model variants

| Dataset | $\mathbf{L}^{(\text{mult})}$ | $\mathbf{S}$ |
|---|---|---|
| | $(\gamma_0, \gamma_1)$ | $(\varepsilon, \beta_1)$ |
| SBM Baseline | (5.0, 0.1) | (0.1, 0.5) |
| SBM Dense-Cliques | (0.1, 0.1) | $(10^{-6}, 0.5)$ |
| SBM Cliques+Noise | (0.1, 0.1) | $(10^{-6}, 0.5)$ |
| SBM Planted-Clique | (0.1, 0.1) | (1.0, 2.0) |
| SBM Small-Dense | (0.1, 1.0) | (1.0, 0.5) |
| SBM Large-Sparse | (1.0, 5.0) | $(10^{-6}, 0.5)$ |
| SBM Unequal-Sizes | (5.0, 5.0) | (0.5, 2.0) |
| SBM Bridge-Only | (0.1, 0.1) | $(10^{-6}, 0.5)$ |

The fundamental solution to this equation is the heat kernel $k_t(x, y)$, which quantifies the amount of heat transferred from node $x$ to node $y$ after time $t$. The Heat Kernel Signature is defined as the diagonal of this kernel:

$$\text{HKS}(x, t) = k_t(x, x) = \sum_{k=0}^{N-1} e^{-\lambda_k t} \phi_k(x)^2, \tag{30}$$

where $\{(\lambda_k, \phi_k)\}_{k=0}^{N-1}$ are the eigenvalue-eigenvector pairs of the Laplacian, ordered by increasing eigenvalue.

**Geometric Interpretation**. The exponential term $e^{-\lambda_k t}$ acts as a time-dependent low-pass filter in the spectral domain. At small time scales $t$, high-frequency eigenmodes (large $\lambda_k$) are rapidly suppressed, emphasizing local geometric structure. Conversely, large time scales preserve only low-frequency components, capturing global topological properties. This multi-scale behavior makes HKS particularly effective for characterizing graph geometry across different resolutions.

**Invariance Properties**. A key advantage of HKS is its invariance to isometric deformations of the underlying space. Since the spectrum of the Laplacian is preserved under isometries, and the signature depends only on squared eigenvector components, HKS is also invariant to sign flips in the eigenbasis.

**Practical Implementation**. In practice, we evaluate Equation 30 at a logarithmically-spaced sequence of time scales $\{t_1, \ldots, t_M\}$ to construct a feature matrix $\mathbf{H} \in \mathbb{R}^{N \times M}$, where each row $\mathbf{H}_{i,:}$ provides a dense, multi-scale descriptor for node $i$. Following Sun et al. (2009), we use $M = 16$ time samples in log scale spacing spanning from 0.1 to 24 with all laplacians.

### B.3. Clustering Performance in the Sparse Regime

As demonstrated in Table 7, collapsed-operator ($\mathbf{S}_\varepsilon$) clustering outperforms standard spectral clustering based on the graph Laplacian ($\mathbf{L}^\mathbf{G}$) in the regime where $q/p \le 0.10$. During the lifting phase, we observe that higher-order cells in this sparse regime contain discriminative structural information that the standard graph Laplacian fails to capture. While the relative utility of this higher-order signal diminishes as $q/p$ increases, these results highlight that "filling in" selected triangles and tetrahedra yields a robust downstream clustering signal when mediated by the collapsed operator $\mathbf{S}_\varepsilon$.

**Table 7.** Clustering performance comparison across varying $q/p$ ratios.

| $q/p$ | $\mathbf{L}^\mathbf{G}$ | $\mathbf{S}_\varepsilon$ | Intra-$\Delta$ % |
|---|---|---|---|
| 0.050 | 0.794 | 0.817 | 97.7 |
| 0.100 | 0.735 | 0.762 | 90.7 |
| 0.500 | 0.454 | 0.393 | 23.3 |
| 0.875 | 0.355 | 0.332 | 7.8 |

## C. Additional Theoretical Results

### C.1. Note on Cheeger Inequalities

The asymmetry between upper and lower bounds is a fundamental obstacle in extending Cheeger inequalities to higher-order structures. For $k$-dimensional simplicial complexes $X$, one can define a combinatorial expansion constant $h(X)$ generalizing the graph Cheeger constant, and relate it to the smallest non-trivial eigenvalue $\lambda(X)$ of the $(k-1)$-dimensional upper

Laplacian. Gundert & Szedlák (2014) proved that the upper bound $\lambda(X) \leq h(X)$ extends to this setting, but showed there is *no higher-dimensional analogue of the lower bound*: one can construct families of simplicial complexes that are spectrally expanding (large $\lambda(X)$) but not combinatorially expanding (small $h(X)$). In other words, spectral and combinatorial notions of expansion decouple in higher dimensions. See also Parzanchevski et al. (2016) for related isoperimetric inequalities on simplicial complexes. Establishing conditions under which lower Cheeger-type bounds hold for operators on higher-order structures (like the collapsed operator $\mathbf{S}$) remains an exciting future research direction.

### C.2. Proof of Positive Semi-definiteness

In this section, we establish conditions under which the regularized Graded Laplacian $\mathbf{L}^\star$ is positive semi-definite (PSD).

**Theorem (3.2 (PSD of Graded Laplacian, restated)).** Let $\mathbf{L}^\star$ be the Graded Laplacian on $C = \bigoplus_{k=0}^{K} C_k$ with adjacency weights $\beta_k \geq 0$ and coupling weights $\gamma_k \geq 0$.

For a matrix $\mathbf{B}$ with singular values $\sigma_1 \geq \cdots \geq \sigma_r > 0 = \sigma_{r+1} = \cdots$, define $\sigma_{\min}^+(\mathbf{B}) := \sigma_r$, with $\sigma_{\min}^+(\mathbf{0}) := +\infty$.

Then $\mathbf{L}^\star \succeq 0$ if

$$\gamma_k \leq \beta_{k+1} \cdot \sigma_{\min}^+(\mathbf{B}_{k+1}) \quad \text{for all } k = 0, \ldots, K-1. \tag{31}$$

*Proof.* Let $\mathbf{x} = (x_0, x_1, \ldots, x_K) \in C$, we proceed by examining the conditions where the quadratic form $Q(\mathbf{x}) = \mathbf{x}^\top \mathbf{L} \mathbf{x} \geq 0$. Note that $Q(\mathbf{x})$ expands as:

$$Q(\mathbf{x}) = \sum_{k=0}^{K} x_k^\top \mathbf{D}_k x_k - 2 \sum_{k=0}^{K-1} \gamma_k \langle x_k, \mathbf{B}_{k+1} x_{k+1} \rangle \tag{32}$$

where $\mathbf{D}_k = \beta_k \mathbf{B}_k^\top \mathbf{B}_k + \beta_{k+1} \mathbf{B}_{k+1} \mathbf{B}_{k+1}^\top$ (with the rank-0 and rank-$K-1$ terms $\mathbf{D}_0 = \beta_0 \mathbf{B}_1 \mathbf{B}_1^\top$ and $\mathbf{D}_K = \beta_{K-1} \mathbf{B}_K^\top \mathbf{B}_K$).

Substituting and regrouping by incidence operator $\mathbf{B}_{k+1}$:

$$Q(\mathbf{x}) = \sum_{k=0}^{K-1} \left[ \beta_{k+1} \| (\mathbf{B}_{k+1})^\top x_k \|^2 + \beta_{k+1} \| \mathbf{B}_{k+1} x_{k+1} \|^2 - 2\gamma_k \langle x_k, \mathbf{B}_{k+1} x_{k+1} \rangle \right] \tag{33}$$

$$= \sum_{k=0}^{K-1} Q_k(x_k, x_{k+1}) \tag{34}$$

where each $Q_k$ couples only adjacent ranks $(k, k+1)$ through $\mathbf{B}_{k+1}$.

Now consider the Singular Value Decomposition (SVD) of $\mathbf{B}_{k+1} = U\Sigma V^\top$ with $\Sigma = \mathrm{diag}(\sigma_1, \ldots, \sigma_r, 0, \ldots)$ and $\sigma_1 \geq \cdots \geq \sigma_r > 0$. Define transformed coordinates $\tilde{x}_k = U^\top x_k$ and $\tilde{x}_{k+1} = V^\top x_{k+1}$. Then:

$$\| \mathbf{B}_{k+1}^\top x_k \|^2 = \| \Sigma \tilde{x}_k \|^2 = \sum_{i=1}^{r} \sigma_i^2 \tilde{x}_{k,i}^2 \tag{35}$$

$$\| \mathbf{B}_{k+1} x_{k+1} \|^2 = \| \Sigma \tilde{x}_{k+1} \|^2 = \sum_{i=1}^{r} \sigma_i^2 \tilde{x}_{k+1,i}^2 \tag{36}$$

$$\langle x_k, \mathbf{B}_{k+1} x_{k+1} \rangle = \langle \tilde{x}_k, \Sigma \tilde{x}_{k+1} \rangle = \sum_{i=1}^{r} \sigma_i \tilde{x}_{k,i} \tilde{x}_{k+1,i} \tag{37}$$

Substituting into $Q_k$:

$$Q_k = \sum_{i=1}^{r} \left[ \beta_k \sigma_i^2 \tilde{x}_{k,i}^2 - 2\gamma_k \sigma_i \tilde{x}_{k,i} \tilde{x}_{k+1,i} + \beta_k \sigma_i^2 \tilde{x}_{k+1,i}^2 \right] \tag{38}$$

$$= \sum_{i=1}^{r} \sigma_i \left[ \beta_k \sigma_i \tilde{x}_{k,i}^2 - 2\gamma_k \tilde{x}_{k,i} \tilde{x}_{k+1,i} + \beta_k \sigma_i \tilde{x}_{k+1,i}^2 \right] \tag{39}$$

$$= \sum_{i=1}^{r} \sigma_i \begin{pmatrix} \tilde{x}_{k,i} \\ \tilde{x}_{k+1,i} \end{pmatrix}^\top \begin{pmatrix} \beta_k \sigma_i & -\gamma_k \\ -\gamma_k & \beta_k \sigma_i \end{pmatrix} \begin{pmatrix} \tilde{x}_{k,i} \\ \tilde{x}_{k+1,i} \end{pmatrix} \tag{40}$$

Each $2 \times 2$ matrix is PSD iff its determinant $\beta_k^2 \sigma_i^2 - \gamma_k^2 \geq 0$, i.e., $\gamma_k \leq \beta_k \sigma_i$. Since this must hold for all $i = 1, \ldots, r$, the binding constraint is at the smallest positive singular value: $\sigma_{\min}^+ = \sigma_r$. □

### C.3. Approximation Error of the Regularized Collapse

*Proof of thm. 3.10.* Write the vertex/higher-order coupling block as $\mathbf{X} = [-\gamma_0 \mathbf{B}_1, \mathbf{0}, \ldots]$. Because $\mathbf{L}^\star \succeq 0$, its principal submatrix $\mathbf{C}$ is positive semidefinite. Let $\mathbf{v} \in \ker(\mathbf{C})$ and write $\mathbf{v} = (\mathbf{v}_1, \mathbf{v}_2, \ldots)$ according to the higher-order ranks. The quadratic form of $\mathbf{C}$ is the quadratic form of $\mathbf{L}^\star$ with the vertex signal set to zero. Using the same nonnegative pairwise decomposition as in the proof of thm. 3.2,

$$
\mathbf{v}^\top \mathbf{C} \mathbf{v} = \beta_1 \|\mathbf{B}_1 \mathbf{v}_1\|^2 + \sum_{k=1}^{K-1} Q_k(\mathbf{v}_k, \mathbf{v}_{k+1}),
$$

where each $Q_k$ is positive semidefinite under the PSD condition. Since $\mathbf{v} \in \ker(\mathbf{C})$, the left-hand side is zero; hence $\beta_1 \|\mathbf{B}_1 \mathbf{v}_1\|^2 = 0$. If $\beta_1 > 0$, this gives $\mathbf{B}_1 \mathbf{v}_1 = 0$. If $\beta_1 = 0$, the PSD condition gives $\gamma_0 = 0$. Thus

$$
\mathbf{X} \mathbf{v} = -\gamma_0 \mathbf{B}_1 \mathbf{v}_1 = 0,
$$

which proves $\ker(\mathbf{C}) \subseteq \ker(\mathbf{X})$.

Let

$$
\mathbf{C} = \sum_{i=1}^r \lambda_i \mathbf{u}_i \mathbf{u}_i^\top, \qquad P_{\ker} = \sum_{i=r+1}^N \mathbf{u}_i \mathbf{u}_i^\top,
$$

where $\lambda_i > 0$ are the positive eigenvalues of $\mathbf{C}$. Then

$$
(\mathbf{C} + \epsilon \mathbf{I})^{-1} = \sum_{i=1}^r \frac{1}{\lambda_i + \epsilon} \mathbf{u}_i \mathbf{u}_i^\top + \frac{1}{\epsilon} P_{\ker}.
$$

Since $\ker(\mathbf{C}) \subseteq \ker(\mathbf{X})$, we have $\mathbf{X} P_{\ker} = 0$. The apparent $\frac{1}{\epsilon}$ kernel term therefore drops out after multiplying by the coupling blocks.

The pseudo-inverse is

$$
\mathbf{C}^\dagger = \sum_{i=1}^r \frac{1}{\lambda_i} \mathbf{u}_i \mathbf{u}_i^\top.
$$

If $r = 0$, then $\mathbf{C} = 0$ and the kernel result implies $\mathbf{X} = 0$, so $\mathbf{S}_\epsilon = \mathbf{S}^\dagger = \mathbf{A}$. Otherwise,

$$
\mathbf{S}_\epsilon - \mathbf{S}^\dagger = \mathbf{X} \left( \mathbf{C}^\dagger - (\mathbf{C} + \epsilon \mathbf{I})^{-1} \right) \mathbf{X}^\top = \mathbf{X} \left( \sum_{i=1}^r \frac{\epsilon}{\lambda_i(\lambda_i + \epsilon)} \mathbf{u}_i \mathbf{u}_i^\top \right) \mathbf{X}^\top.
$$

Taking operator norms gives

$$
\|\mathbf{S}_\epsilon - \mathbf{S}^\dagger\|_2 \leq \|\mathbf{X}\|_2^2 \max_{1 \leq i \leq r} \frac{\epsilon}{\lambda_i(\lambda_i + \epsilon)} \leq \frac{\epsilon \|\mathbf{X}\|_2^2}{\lambda_+(\lambda_+ + \epsilon)},
$$

with $\lambda_+ = \lambda_{\min}^+(\mathbf{C})$. This proves the claimed convergence and the stated error bound. □

### C.4. Block Diagonalization of the Graded Laplacian

In this section, we formally prove that the Graded Laplacian $\mathbf{L}^*$ decomposes into independent blocks corresponding to disconnected components. For CW complexes, the operators are the cellular boundary matrices. For the incidence-based construction, the same statement applies to finite graded posets equipped with adjacent-rank incidence matrices.

**Theorem C.1.** *Let $C$ be either a finite CW complex with cellular boundary matrices or a finite graded poset with adjacent-rank incidence matrices. Suppose $C$ consists of $m$ disjoint connected components denoted by $C_1, C_2, \ldots, C_m$. Let $\mathbf{L}^* \in \mathbb{R}^{N \times N}$ be the Graded Laplacian of $C$, where indices are initially ordered by rank.*

*There exists a permutation matrix $\mathbf{P} \in \{0, 1\}^{N \times N}$ such that the permuted Laplacian is block diagonal:*

$$\mathbf{P}\mathbf{L}^*\mathbf{P}^\top = \begin{pmatrix} \mathbf{L}^*_{C_1} & \mathbf{0} & \dots & \mathbf{0} \\ \mathbf{0} & \mathbf{L}^*_{C_2} & \dots & \mathbf{0} \\ \vdots & \vdots & \ddots & \vdots \\ \mathbf{0} & \mathbf{0} & \dots & \mathbf{L}^*_{C_m} \end{pmatrix} \tag{41}$$

*where $\mathbf{L}^*_{C_i}$ is the Graded Laplacian restricted to the i-th connected component.*

*Proof.* We proceed by structural induction on the number of connected components $m$.

**Base Case ($m = 1$):** Consider $C$ with exactly one connected component $C_1$. In this case, the Graded Laplacian $\mathbf{L}^*$ is already the operator for the single component $C_1$. The required permutation matrix is simply the identity matrix $\mathbf{P} = \mathbf{I}_N$. The theorem holds trivially.

**Inductive Step:** Assume the theorem holds for any such structure with $k$ connected components. We show it holds for $C$ with $k + 1$ connected components, denoted $C_1, \dots, C_{k+1}$.

We can partition $C$ into two disjoint sets: the first component $C_1$ and the union of the remaining components $C' = \bigcup_{j=2}^{k+1} C_j$.

Let $I$ be the set of all cell indices in $C$. We partition $I$ into $I_1$ (indices of cells in $C_1$) and $I'$ (indices of cells in $C'$). By definition of connected components, there are no boundary or incidence relations between $C_1$ and $C'$. Consequently, any entry in $\mathbf{L}^*$ correlating an index from $I_1$ with an index from $I'$ is zero.

Let $\mathbf{P}_1$ be a permutation matrix that reorders the indices such that all $i \in I_1$ appear before all $j \in I'$. Applying this similarity transformation yields:

$$\mathbf{P}_1\mathbf{L}^*\mathbf{P}_1^\top = \begin{pmatrix} \mathbf{L}^*_{C_1} & \mathbf{0} \\ \mathbf{0} & \mathbf{L}^*_{C'} \end{pmatrix} \tag{42}$$

Here, $\mathbf{L}^*_{C_1}$ is the Graded Laplacian of the first component, and $\mathbf{L}^*_{C'}$ is the Graded Laplacian of the remaining sub-complex.

Since $C'$ consists of exactly $k$ connected components ($C_2, \dots, C_{k+1}$), we invoke the inductive hypothesis. There exists a permutation $\mathbf{P}'$ such that:

$$\mathbf{P}'\mathbf{L}^*_{C'}(\mathbf{P}')^\top = \text{diag}(\mathbf{L}^*_{C_2}, \dots, \mathbf{L}^*_{C_{k+1}}) \tag{43}$$

We construct the full permutation $\mathbf{P}$ by combining the identity for the first block with $\mathbf{P}'$ for the second block:

$$\mathbf{P} = \begin{pmatrix} \mathbf{I}_{|I_1|} & \mathbf{0} \\ \mathbf{0} & \mathbf{P}' \end{pmatrix} \mathbf{P}_1 \tag{44}$$

Applying this to the original matrix yields the full block diagonal structure:

$$\mathbf{P}\mathbf{L}^*\mathbf{P}^\top = \text{diag}(\mathbf{L}^*_{C_1}, \mathbf{L}^*_{C_2}, \dots, \mathbf{L}^*_{C_{k+1}}) \tag{45}$$

Thus, by the principle of induction, the theorem holds for all $m \geq 1$. □

## D. Extended Related Work

**Schur Complement Methods.** The Schur complement plays a central role in efficient graph algorithms. It enables graph sparsification while preserving spectral properties (Durfee et al., 2017), facilitates effective resistance estimation (Dorfler & Bullo, 2010), and provides the theoretical foundation for approximate Gaussian elimination on graph Laplacians (Kyng & Sachdeva, 2016). A common theme across these applications is the elimination of a spatial subset of vertices while maintaining key spectral quantities, which parallels our approach to boundary operators. In our work we don't sparsify the vertex set but capture the higher order dynamics and collapsing them onto the vertex space.

**Graph Kernels and Gaussian Processes.** Laplacian based kernel methods on graphs have been extensively studied, particularly for modeling diffusion processes. Heat kernels (Bai & Hancock, 2004) and Matérn kernels (Borovitskiy et al., 2021) provide principled approaches to defining similarity on graph-structured domains. (Shuman et al., 2013) provides theoretical foundations for signal processing in the graph domain.

**Table 8.** Runtime scaling with increasing higher-order topological density for a fixed vertex count ($V = 400$).

| Vertices | Edges | Triangles | Tetrahedra | Implicit Sparse CG (s) |
|---|---|---|---|---|
| 400 | 696 | 18 | 0 | 0.50 |
| 400 | 1650 | 296 | 4 | 12.43 |
| 400 | 2081 | 625 | 12 | 36.32 |
| 400 | 2702 | 1416 | 71 | 95.73 |
| 400 | 4159 | 5504 | 1148 | 223.93 |

**Table 9.** Runtime scaling comparison in the sparse regime.

| Vertices | Edges | Triangles | Tetrahedra | Sparse Direct Fact. (s) | Implicit Sparse CG [Ours] (s) |
|---|---|---|---|---|---|
| 200 | 742 | 240 | 9 | 0.08 | 5.41 |
| 800 | 4434 | 741 | 2 | 4.58 | 20.87 |
| 1200 | 7311 | 939 | 7 | 33.59 | 43.96 |
| 2000 | 13840 | 1180 | 1 | 323.14 | 75.51 |
| 3000 | 23297 | 1571 | 1 | OOM | 404.77 |

**Higher-Order Message Passing**. Recent work has extended message passing beyond pairwise interactions. Higher-order message passing (HOMP) frameworks leverage simplicial complexes to capture multi-way relationships (Battiloro et al., 2025; Roddenberry et al., 2021). Related efforts inject topological priors directly into message-passing neural networks (Horn et al., 2022; Chen et al., 2021), extending their expressiveness

## E. Computational Scalability and the Implicit Operator $\mathbf{S}_\epsilon$

In applications such as spectral clustering, the dense collapsed operator $\mathbf{S}_\epsilon$ does not need to be constructed explicitly. Instead, we employ a Krylov subspace approach that requires only matrix-vector multiplications of the form $\mathbf{v}_{\text{next}} = \mathbf{S}_\epsilon \mathbf{v}$. Specifically, this is evaluated as:

$$\mathbf{S}_\epsilon \mathbf{v} = \mathbf{A}\mathbf{v} - \mathbf{X}(\mathbf{C} + \epsilon\mathbf{I})^{-1}\mathbf{X}^\top\mathbf{v} \tag{46}$$

Evaluating this product avoids dense matrix inversion and only requires solving the linear system $(\mathbf{C} + \epsilon\mathbf{I})\mathbf{z} = \mathbf{y}$ via the Conjugate Gradient (CG) method.

A single matrix-vector multiplication with the sparse block $C$ takes $O(M)$ operations, where $M$ is the total number of non-zero entries (incidences) in the higher-order complex. Therefore, the theoretical runtime scales linearly as $O(N_{CG} \cdot M)$, where $N_{CG}$ is the number of CG iterations.

To empirically validate this scaling behavior, we conducted an experimental analysis measuring runtime as higher-order topological density increases. As shown in Table 8, the runtime of our implicit sparse CG approach scales predictably with the number of higher-order structures.

It is worth noting that the Conjugate Gradient method is inherently sensitive to matrix conditioning. Consequently, higher-order incidences may require more iterations to converge if they worsen the operator's conditioning. Furthermore, increased density affects the cost per iteration through more expensive matrix-vector products.

Nevertheless, when the regularized system retains a favorable sparse structure, CG remains practical and highly preferable to direct solvers for large-scale problems. Table 9 demonstrates this advantage by comparing the runtime of our implicit CG method against a standard sparse direct factorization solver across increasingly large complexes.

As highlighted in Table 9, the implicit CG formulation rapidly overtakes the direct solver as the problem size scales. At $V = 2000$, our implementation completes the operation more than 4× faster than the direct sparse solver (75.51s versus 323.14s). Furthermore, the memory efficiency of the Krylov subspace approach allows it to successfully process graphs at $V = 3000$, whereas the direct solver fails due to Out-Of-Memory (OOM) errors.

