# OpenReview forum: "Collapsed Effective Operators for Higher-order Structures"
_ICML.cc/2026/Conference — ICML 2026 regular_

### Official Review · Reviewer_jTGk · 2026-02-25

**Soundness:** 2
**Presentation:** 3
**Significance:** 2
**Originality:** 2
**Overall Recommendation:** 4
**Confidence:** 4

**Summary:**

This paper introduces Collapsed Effective Operators, a novel framework for incorporating higher-order topological structures into vertex-level learning tasks. The key idea is to marginalize higher-order cells onto the vertex set via the Schur complement of a graded Laplacian, yielding a dense, positive semi-definite operator that encodes long-range interactions induced by topology. The authors provide theoretical guarantees, including spectral bounds and energy interpretations, and demonstrate empirical improvements in spectral clustering, protein structure segmentation, graph classification, and topological feature extraction. The method is shown to be computationally efficient compared to existing higher-order message-passing architectures.

**Compliance With Llm Reviewing Policy:**

Affirmed.

**Final Justification:**

Thanks for your response. The rebuttal basically addressed my own concerns, however i dont think I can raise my score from weak accept to accept.

**Key Questions For Authors:**

1. The related work section should incorporate and discuss recently published key studies to ensure the manuscript reflects the latest research advances.
2. The authors should carefully proofread this paper and correct all the typos in the revision. In the current version, there are some typos/grammar errors. For example, “computational improvmeents” on lines 404-405.
3. The analysis of the tabular results in the experimental section merely limits itself to numerical comparisons, and it is recommended that the authors add an interpretation of the underlying reasons.
4. The baseline methods used in the experiments are not sufficiently up-to-date. It is recommended to include and compare against more recent SOTA methods.
5. The scale of datasets used in the existing experiments is relatively limited, and there is a lack of validation on large-scale higher-order complexes. It is recommended that the authors supplement experiments on large-scale synthetic higher-order complexes to verify the performance and efficiency of the method on large-scale data.

**Limitations:**

The approach approximates the full higher-order spectrum during collapse, potentially losing some topological details. The resulting operator may be dense, departing from the sparsity of traditional Laplacians, though this reflects mediated non-local couplings. Current evaluations are on static structures; dynamic or very large-scale applications may require further optimizations.

**Strengths And Weaknesses:**

S1: The proposed operator provides a principled and unified way to aggregate higher-order topological information into vertex-level representations, addressing a key limitation in topological deep learning.
S2: Empirical validation is comprehensive, covering synthetic benchmarks, real-world networks, protein data, and molecular classification tasks.
S3: The method is computationally scalable via iterative solvers and avoids the dense operator bottleneck.

W1: The collapsed operator can become dense, potentially increasing computational overhead for very large graphs despite sparsity-preserving techniques.
W2: Empirical evaluations focus on specific benchmarks; broader testing on larger-scale or dynamic topologies could strengthen generalizability claims.

---

> ### Author Rebuttal · Authors · 2026-03-31
>
> We thank the reviewer for the positive and constructive feedback. We’re glad they found our method principled and the evaluation comprehensive. In the following, we would like to address their comments.
>
> **W1** (abstract): We believe our abstract is already quite concise at less than 20 lines. Moreover, it should be self-contained as it will often be read in isolation without the full introduction at hand. If the reviewer insists on shortening it, we would kindly propose specific modifications during the author-reviewer discussion phase.
>
> **W2, Q2, Q3** (contrastive learning): To clarify, our paper doesn’t deal with graph contrastive learning, vertex entanglement, or fuzzy logic. Because these comments appear to describe a different manuscript, we suspect there may have been a mix-up. As a result, we are unable to address the specific weaknesses raised, but we remain fully available throughout the discussion period to answer any questions regarding our actual submission.
>
> **Q1**. Could the reviewer specify additional related works we should include?
>
> We are happy to clarify any further questions during the discussion period.

---

> > ### Author Rebuttal · Reviewer_jTGk · 2026-04-01
> >
> > Thank you to the authors for their response. However, the issues regarding comparing with the latest baseline methods and incorporating the latest relevant works (e.g., those from 2025) have not yet been effectively resolved. Therefore, I will maintain my original score.

---

> > > ### Author Response · Authors · 2026-04-02
> > >
> > > Thank you for the updated review; we understand such mixups can happen. Please find a response to your new points below.
> > >
> > > **W1 potential density**
> > >
> > > Reviewer MmFQ asked a related question in their W1. In the response, we show that with our proposed Alg 1, the CG solver can leverage the sparsity of C, enabling scalability to thousands of vertices without materializing the dense operator. To empirically substantiate scalability, we refer to the runtime tables added in our response to MmFQ W1.
> > >
> > > **W2: Broader Empirical Evaluations**
> > > We evaluated our method across a broad range of tasks, including signal smoothing, clustering on real and synthetic benchmarks, regression of simplicial complex properties, and protein structure segmentation. In the rebuttal, we additionally conduct a node classification experiment to demonstrate the value of our operator as a positional encoder for neural networks (see our response to fk2w W2). Specifically, we derived positional encodings from the graph Laplacian and the Schur collapse operator, then evaluated them in a semi-supervised transductive setting with established learning architectures.
> > >
> > > Regarding dynamic topology (topological changes), Tab. 1 (Sec. 4.1) demonstrates robustness to structural perturbations. We progressively degrade a random geometric simplicial complex by injecting 0–200 shortcut cells, thereby increasing topological noise as the complex structure varies. The collapsed operator proves substantially more robust than both the graph Laplacian and the multiorder Laplacian. Furthermore, in our response to Reviewer MmFQ (W1), we measure computation time for a progressively densifying complex. As the number of higher-order cells increases, the cost of applying the Schur complement via the CG solver grows accordingly, which is an inherent trade-off of encoding richer topological structure at the vertex level.
> > >
> > > **Q4 & Q1 Missing comparison to SOTA and related work**
> > > Our main contribution is a novel operator that effectively marginalizes higher-order topology to the vertex set. The Multiorder Laplacian (presented in Def. 2.9), against which we compare throughout our experiments, is applicable to an analogous problem setting. However, other operators, such as the Hodge Laplacian or the normalized Hodge 1-Laplacian (Schaub et al., 2020), can only be analyzed on isolated ranks. While the Dirac operator (Calmon et al., 2021) operates on the full complex, it cannot be natively marginalized to a specific rank.
> > >
> > > We also reviewed the recent literature and expanded our related work section. Specifically, we incorporate two references that use the Schur complement to reduce graph operators. Dörfler & Bullo [1] introduce Kron reduction, in which the Schur complement eliminates a spatial subset of nodes to produce an equivalent reduced graph on the remaining vertices. While this construction reduces within a single rank, we use the Schur complement to collapse across ranks, marginalizing higher-order cells into effective vertex-level couplings that do not exist in any subgraph. Mémoli et al. [2] use the Schur complement to define a persistent Laplacian relating a subcomplex K to a different "ambient" complex L. In contrast, our operator acts on a single complex by integrating the full graded Laplacian into a single vertex-level operator that encodes how higher-order topology mediates node dynamics.
> > >
> > > [1] Dörfler et al. "Kron reduction of graphs with applications to electrical networks." IEEE Transactions on Circuits and Systems I: Regular Papers, 2012
> > >
> > > [2] Mémoli et al. "Persistent Laplacians: Properties, algorithms and implications." SIAM Journal on Mathematics of Data Science, 2022
> > >
> > > **Q2: Typos**
> > > We thank the reviewer for spotting this typo. We have identified several more and also corrected them:
> > >
> > > - Page 3, line ~135: "in where hierarchical closure is not preserved" → "in which hierarchical closure..."
> > > - Page 8, line ~397: "SMNCN" → "SMCN"
> > > - Page 8, line ~405: "improvmeents" → "improvements"
> > > - Page 8, line ~425: "lapcian" → "Laplacian".
> > >
> > > **Q3 Improved experimental interpretation**
> > > We have improved interpretive discussion in the experimental section, clarifying, for instance, when rank-local signals favor Hodge-based methods over our collapsed operator (see also our response to fk2w W3 for details). In brief, our operator excels when higher-order topology carries a discriminative signal for vertex-level tasks (we demonstrate this with protein secondary/tertiary structure), whereas rank-wise methods are better suited when tasks depend on rank-local invariants such as individual Betti numbers.
> > >
> > > **Q5 Limited scalability experiments**
> > > We kindly refer the reviewer to the runtime analysis presented above (W1) and in our response to MmFQ W1, which includes both density-scaling and vertex-scaling experiments.
> > >
> > > We hope these clarifications and additional experiments address the reviewer's concerns. We remain available throughout the discussion period to answer any further questions.

---

### Official Review · Reviewer_3EWJ · 2026-03-12

**Soundness:** 2
**Presentation:** 1
**Significance:** 3
**Originality:** 3
**Overall Recommendation:** 3
**Confidence:** 3

**Summary:**

This paper proposes "Collapsed Effective Operators", a novel mathematical framework designed to address the rank-fusion bottleneck in Topological Deep Learning. Using Schur complementation to a graded Laplacian, the authors marginalize higher-order topological structures (such as simplicial and combinatorial complexes) into a single vertex-level operator. This approach aims to directly embed long-range, topology-mediated interactions into node-level dynamics, offering a principled and streamlined alternative to complex, heuristic-driven higher-order message-passing architectures.

**Compliance With Llm Reviewing Policy:**

Affirmed.

**Final Justification:**

Based on the rebuttal, I am raising the score to weak accept.
The mistakes in the original were not acceptable as a paper for publication, and this is the best score that I can provide.
If the paper gets accepted, I hope that there are no more issues that confuses readers, or if rejected, there should be no such errors so that the contribution of this work is properly validated especially when dealing with the theoretical aspect of ML.

**Key Questions For Authors:**

- Please refer to the detailed points raised in the Weaknesses section.

- Overall, the manuscript contains extensive mathematical notations and specialized concepts borrowed from other domains (e.g., physics and advanced topology). This makes the paper quite difficult for a general audience to follow and highly susceptible to notational or conceptual errors.

- Additionally, given the strong emphasis on the necessity of "node-level" representations in the introduction, it would have been better to include standard node-level tasks, such as node classification, in the empirical analysis to directly validate this core motivation.

**Limitations:**

yes

**Strengths And Weaknesses:**

Strength:
- The primary strength of this work lies in its elegant and highly intuitive mathematical motivation, which draws inspiration from effective theories in physics to simplify multi-rank interactions.
- The proposed single-operator formulation is novel, and it successfully demonstrates an empirical evidence to capture non-local topological motifs, such as $\alpha$-helices in biochemical networks, without the need for heuristic architectural engineering.

Weakness:
- Critical Errors in Notation and Definitions (Sec. 2.1): The incidence matrix $\mathbf{E}$ and boundary operators $\mathbf{D}_k$ are strictly defined as binary matrices over $\{0, 1\}$, neglecting the orientation; which is mathematically invalid as these matrices must be defined over $\{-1, 0, 1\}$. This results to critical error on definition of graph Laplacian $\mathcal{L}^G=EE^T$ (i.e., it results signless graph Laplacian ($\mathbf{L}^G = \mathbf{D} + \mathbf{A}$) rather than the standard graph Laplacian ($\mathbf{L}^G = \mathbf{D} - \mathbf{A}$)) and the property Eq.(1).

- Inconsistent Notation and Dimensional Mismatch (Sec. 2.2): The notation for the boundary matrix $\mathbf{D}_k$ is highly inconsistent. It is defined in Definition 2.2, but in Remark 2.8, it is redefined differently.

- Additionally, the text in Remark 2.8 incorrectly states that $\partial_k$ has the matrix representation $\mathbf{D}_k^\top$. Given  $\mathbf{D}_k$,
the correct matrix representation for the boundary operator

$\partial_k : C_k \to C_{k-1}$
is simply $\mathbf{D}_k$, not its transpose.

- Reversed Physical Interpretation of Effective Operators (Sec. 3.2): Equation (14) seems fundamentally incorrect. The variational form $\min_{\mathbf{x}: x_i - x_j = 1} \mathbf{x}^\top \mathbf{L} \mathbf{x}$ computes the effective conductance, not the effective resistance (which requires the pseudo-inverse $\mathbf{L}^+$). Consequently, the proof in Corollary 3.8 and the subsequent physical analysis are entirely backward. The authors must address this fundamental contradiction.

- The citation of Fig. 1 in the Introduction (mentioning "all require vertex-level representations") misaligns with the figure's actual content, which demonstrates the superiority of the proposed operator. Furthermore, the caption of Fig. 1 lacks sufficient context for a general audience to understand why capturing such topological clusters (e.g., the $\alpha$-helix) is important compared to other nodes.

- The conceptual bridge explaining "why global topological structures are essential for local node-level tasks" could be strengthened. This needs clarification in two ways; why is the inclusion of higher-order (often macro or global) structures intrinsically important for local, node-level predictions? If the proposed method successfully projects topology onto the node level, what specific advantages does it hold over existing advanced node-level representation methods?

---

> ### Author Rebuttal · Authors · 2026-03-31
>
> We thank the reviewer for their thoughtful review. We are encouraged that they find our mathematical motivation “elegant and highly intuitive”, the proposed single-operator formulation “novel”, and for recognising that it “successfully demonstrates empirical evidence to capture non-local topological motifs”. In the following, we address all of their comments.
>
> **W1–W3. Errors in Notation and Definitions (Sec. 2.1–2.2).**
>
> We sincerely thank the reviewer for pointing out these notational issues. They stem from generalizing the theory from combinatorial complexes, which do not require signed boundary matrices, to simplicial / CW complexes, where we unfortunately failed to update the notation. We have corrected these errors in the manuscript. We emphasize that none of these notational corrections affect the operator construction or the PSD/Schur-complement analysis that follows; only the interpretation of Eq. (14) and Cor. 3.8 changes, as discussed in W4. The corrections are:
>
> 1. **Incidence and boundary matrices (W1).** The incidence matrix **$E$** and boundary operators **$D_k$** are now correctly defined with entries in {−1, 0, 1}, restoring the orientation. Consequently, the graph Laplacian is properly stated as **$L^G = D − A$** (not $D + A$), and Eq. (1) holds as intended.
> 2. **Consistent notation for $D_k$ (W2).** We have unified the notation for **$D_k$** across Definition 2.2 and Remark 2.8, ensuring dimensional consistency.
> 3. **Matrix representation of $∂_k$ (W3).** We corrected Remark 2.8: the matrix representation of the boundary operator **$∂_k$** **$:C_k → C_{k−1}$** is **$D_k$**, not its transpose. The Hodge Laplacian in Eq. (4) now reads **$Δ_k = D_{k}^⊤ D_{k} + D_{k+1} D_{k+1}^⊤$** with consistent notation throughout.
>
> From Proposition 3.1 onward, the derivations rely mainly on block PSD properties and singular values. In Appendix C.2., the singular-value argument is unchanged once the sign convention is made consistent. Proposition 3.6 and Cor. 3.7 use only the PSD property of the Schur complement and the PD/PSD properties of the graded-Laplacian blocks, which remain valid with signed boundary matrices. Likewise, Cor. 3.8 remains valid under the revised interpretation in W4, as it only uses the PSD properties of the Schur collapse and the upper block matrix.
>
> **W4. Reversed Physical Interpretation of Effective Operators (Sec. 3.2).**
>
> We thank the reviewer for identifying this. Indeed, the variational quantity $\min_{x: x_i − x_j = 1} x^\top L x$ computes the effective conductance (the reciprocal of the effective resistance), not the effective resistance itself. We have corrected the labelling in Eq. (14) and revised Corollary 3.8 accordingly. Importantly, the matrix inequality and the corollary’s proof are unchanged: the spectral bound S ⪯ A from Proposition 3.6 still holds, and the proof of the corollary proceeds in the same way, just with a different subsequent interpretation. In particular, the result implies that higher-order cells lower the variational conductance between common vertices, and therefore increase the effective resistance rather than lowering it as previously stated. We also updated the discussion in Sec. 3.2 to reflect this correction.
>
> **W5. Misalignment of Fig. 1 Citation and Insufficient Caption.**
>
> Indeed, Fig 1 can be better contextualised in the main text, with a more descriptive caption. We have now placed the reference to Fig. 1 on line 86, after we introduce our operator. The caption now reads: “Our proposed collapsed effective operator (right) marginalises higher-order information into a single vertex-level operator. This enables node-level clustering that is aware of higher-order topological structures in proteins (the $\alpha$-helix, shown in turquoise), whereas the graph Laplacian (left) is agnostic to them.”
>
> **W6. Clarifying the main message**
>
> We now better incorporate these conceptual bridges in the paper:
>
> Why global topology matters. Node-level tasks such as classification and clustering act on vertex signals, but the underlying interactions are often higher-order, e.g., protein secondary/tertiary structure. Observing only the primary structure (the amino-acid backbone) can miss this non-local context. By collapsing higher-order topology onto vertices via the Schur complement, our operator incorporates information that rank-wise Laplacians do not encode directly in a single vertex-level operator.
>
> Advantages over existing node-level methods. Graph Laplacians capture pairwise vertex interactions, while higher-order methods typically rely either on message passing, which can be computationally expensive, or on additive aggregation such as the Multi-Laplacian, which does not couple ranks. Our method addresses this gap with a principled operator that combines multi-rank interactions.
>
> We also add node-classification experiments using the spectrum of $L^\star$ as positional encodings for neural networks; see fk2w (W2).

---

> > ### Author Rebuttal · Reviewer_3EWJ · 2026-04-03
> >
> > Thanks for the rebuttal.
> > I am going to adjust my score accordingly, but these issues should not have been there in the original submission as they will lead readers to nowhere.

---

> > > ### Author Response · Authors · 2026-04-06
> > >
> > > We thank the reviewer for taking the time to review our rebuttal and acknowledging that their concerns are fully resolved.
> > >
> > > We would like to clarify that all the issues the reviewer raised have been carefully addressed and fully resolved in the revised version of the paper. While we agree that these issues should ideally not have appeared in the original submission, we believe their presence does not diminish the fact that they have now been comprehensively corrected. This is why a two-stage review involving a revision process exists.
> > >
> > > We appreciate that the reviewer recognized this by marking their concerns as fully resolved. Given this, we kindly encourage the reviewer to consider reflecting this resolution with a positive rating. Alternatively, if there are any remaining concerns, however minor, we would be more than happy to address them promptly. We thank the reviewer again for their constructive feedback and engagement.

---

### Official Review · Reviewer_fk2w · 2026-03-14

**Soundness:** 3
**Presentation:** 3
**Significance:** 3
**Originality:** 3
**Overall Recommendation:** 4
**Confidence:** 3

**Summary:**

The paper introduces Collapsed Effective Operators, where vertex-level operators obtained by marginalizing higher-order topological cells (edges, triangles, etc.) via the Schur complement of a graded Laplacian. The idea is to avoid the rank-by-rank fusion problem that plagues Hodge Laplacians. Instead of computing separate operators at each rank and then figuring out how to combine them, the construction integrates out higher-order cells and delivers a single PSD operator on nodes. The theoretical analysis shows eigenvalue compression, effective resistance reduction, and applicability to simplicial, CW, and combinatorial complexes.

**Compliance With Llm Reviewing Policy:**

Affirmed.

**Final Justification:**

After discussing with the authors, I still consider this paper to be marginally above the bar of acceptance. Hence, my score remains at 4 (weak accept).

**Key Questions For Authors:**

See weaknesses.

**Limitations:**

Yes.

**Strengths And Weaknesses:**

Strengths:

1. The core mathematical idea of this paper is novel. Using Schur complementation to collapse higher-order cells is a natural and principled construction, and the paper does a good job connecting it to physical intuition. Higher-order cells act as "parallel conductance pathways" that lower effective resistance between vertices sharing common cells.

2. The protein segmentation experiment is the highlight of the empirical section. The graph Laplacian clusters by spatial proximity while the collapsed operator recovers the interleaved secondary structure motifs. This qualitative result demonstrates behavior that the graph Laplacian cannot capture.

Weaknesses:

1. There is a gap between the theoretical object and the implemented operator. While the analysis focuses on the Schur complement $S = A - BC^{-1}B^\top$, the algorithm uses the regularized form $S_\varepsilon = A - B(C + \varepsilon I)^{-1}B^\top$. The paper does not analyze the approximation error introduced by this regularization or discuss the sensitivity of downstream spectral quantities to the choice of $\varepsilon$.

2. The experimental evaluation beyond clustering is somewhat limited. The GIFFLAR benchmark (Table 5) is a specialized glycan dataset, and the results there are mixed. Evaluation on at least one widely used molecular benchmark such as QM9 or ZINC would strengthen the empirical evidence.

3. The empirical results provide mixed evidence for the proposed operator. On the Mantra benchmark, the Multiorder Laplacian slightly outperforms the collapsed operator on both $\beta_1$ and $\beta_2$. Similarly, on the GIFFLAR benchmark the results are competitive but not clearly superior to existing methods. The paper would benefit from a clearer discussion of the regimes in which the collapsed operator is expected to outperform rank-wise approaches.

4. The scalability claims are not empirically substantiated. The method relies on solving linear systems involving $C + \varepsilon I$ using conjugate gradients, yet the paper does not report runtime measurements, CG iteration counts, or scaling behavior with respect to the number of higher-order cells.

---

> ### Author Rebuttal · Authors · 2026-03-31
>
> We sincerely thank the reviewer for their constructive and encouraging feedback. We are pleased that the core mathematical approach of using Schur complementation to collapse higher-order cells was found novel and principled. In the following, we would like to address the mentioned critiques and questions.
>
> **W1 Approximation error of the regularisation**
>
> Note that $L_{\star} \succeq 0 \implies C \succeq 0$ as the latter is a principal submatrix. Now consider the eigendecomposition:
> $$ C = \sum_{i=1}^{r} \lambda_i \, u_i u_i^\top + \sum_{i=r+1}^{N} \lambda_i \, u_i u_i^\top,$$
> where $\lambda_1 \geq \cdots \geq \lambda_r > 0 = \lambda_{r+1} = \cdots = \lambda_N$ and $r = \operatorname{rank}(C)$.
> Let $P_{\ker} = \sum_{i=r+1}^{N} u_i u_i^\top$ denote the orthogonal projector onto $\ker(C)$. The regularized inverse used in Algorithm 1 then takes the form:
> $$(C + \epsilon I)^{-1} = \sum_{i=1}^{r} \frac{1}{\lambda_i + \epsilon}\, u_i u_i^\top
>     \\; + \\; \frac{1}{\epsilon}\, P_{\ker}.$$
> This reveals two distinct effects of the regularization. On the positive part of the spectrum, each eigenvalue $\lambda_i$ is shifted to $\lambda_i + \epsilon$, decreasing its contribution by a factor $\lambda_i/(\lambda_i + \epsilon)$. For $\epsilon \ll \lambda_r = \lambda_{\min}^+(C)$, this damping is negligible. On the kernel, however, the regularization materializes eigenvalues $1/\epsilon$, which could dominate the operator $S_\epsilon$ through the term $B(C+\epsilon I)^{-1}B^\top$ when $\epsilon$ is small.
>
> We now argue that the kernel contribution vanishes. Since the coupling matrix $B = [-\gamma_0 B_1,\, 0,\, \ldots]$, we have $Bv = -\gamma_0 B_1 v_1$ for any higher-order signal $v = (v_1, v_2, \ldots)$. If $v \in \ker(C)$, then $v^\top C v = 0$, and since $C$ contains $\beta_0 B_1^\top B_1$ in its top-left block, this gives $\|B_1 v_1\|^2 = 0$, hence $Bv = 0$. Thus $\ker(C) \subseteq \ker(B)$, so $BP_{\ker} = 0$, and the $1/\epsilon$ term drops out of $B(C+\epsilon I)^{-1}B^\top$ entirely. What remains is only the damping on the positive eigenvalues.
>
> We have formalized this for the paper.
>
> **W2: Suggested additional experiments**
>
> While QM9 and ZINC are widely used for graph-level regression tasks, the added benefit of higher-order information has been observed to be limited [1].
> To further demonstrate that our operator captures meaningful higher-order interactions, we construct a node classification experiment on proteins (combinatorial complex as in Sec. 4.2), classifying amino acids by molecular position and acid type using only positional and relational features. We derive positional encodings from the graph Laplacian and the Schur collapse operator, then evaluate in a semi-supervised transductive setting using a GraphTransformer [2] and GCN [3] on two label sets: *Contact* (3 structural classes) and *ResType* (23 amino-acid-based classes).
> | Model+PE  | Contact | ResType |
> | :--- | :--- | :--- |
> | **GCN** + NoPE   | 0.471 ± 0.013 | 0.524 ± 0.019 |
> | **GCN** + LaplacePE  | 0.480 ± 0.010 | 0.582 ± 0.033 |
> | **GCN** + SchurPE   | **0.494 ± 0.013** | **0.666 ± 0.081**|
> | | | |
> | **GT** + NoPE   | 0.463 ± 0.034 | 0.773 ± 0.019 |
> | **GT** + LaplacePE  | 0.551 ± 0.004 |0.762 ± 0.028|
> | **GT** + SchurPE  | **0.573 ± 0.009** | **0.789 ± 0.019**|
>
> Positional encodings derived from $L^\star$  consistently improve performance across both architectures, confirming that the operator provides useful vertex-level signals beyond clustering.
>
> [1] Expressivity and Generalization: Fragment-Biases for Molecular GNNs, Wollenschlaeger et al. ICML 2024
>
> [2] Masked Label Prediction: Unified Message Passing Model for Semi-Supervised Classification", Shi et al. IJCAI 2021
>
> [3] Semi-Supervised Classification with Graph Convolutional Networks, Kipf et al. ICLR 2017
>
> **W3. Mixed performance for graph-level tasks**
>
> We thank the reviewer for this observation and have added a discussion to the experimental section. Our collapsed operator is designed for a regime in which higher-order topological structure carries a discriminative signal for vertex-level tasks, such as tertiary structure in proteins. In this regime, our operator outperforms the graph Laplacian by explicitly aggregating cross-rank interactions into a vertex-level representation, as we have shown in sec 4.2 and 4.3.
>
> The trade-off is that collapsing across ranks does not fully preserve rank-local topological invariants such as individual Betti numbers. On benchmarks like MANTRA, where tasks depend on rank-local signals, rank-wise methods that retain the full Hodge structure are naturally better suited. Nevertheless, our MANTRA experiments show that the collapsed operator retains substantially more topological information than the graph Laplacian.
>
> **W4 Empirical Scalability Claims**
>
> We additionally added scalability measures (see response to **MmFQ** W1).
>
> We hope the clarification and additional experiment are helpful, and we are happy to answer any remaining questions.

---

> > ### Author Rebuttal · Reviewer_fk2w · 2026-04-02
> >
> > I thank the authors for the detailed responses, which address my concerns. Going over everyone's reviews and the responses, it looks like the authors had to change some critical errors in notation and definition during this rebuttal period. This is not ideal. Under this light, I will neither lower nor raise my score at this time.

---

> > > ### Author Response · Authors · 2026-04-02
> > >
> > > We thank the reviewer for considering our responses and glad to hear that our rebuttal addresses their concerns. We would like to emphasize that our notational revisions do not affect the critical parts of our exposition. They relate to the presentation and interpretation, rather than methodological and technical correctness. In particular, all operator constructions, as well as the PSD and Schur-complement analyses, remain fully valid and unchanged. Thanks to the feedback of all the reviewers, with the presented updates, our manuscript is now consistent and even stronger. In light of these improvements as well as the additional experimentation and theoretical analysis we presented to the reviewer, we respectfully invite them to reconsider their final assessment.

---

### Official Review · Reviewer_MmFQ · 2026-03-14

**Soundness:** 3
**Presentation:** 4
**Significance:** 4
**Originality:** 3
**Overall Recommendation:** 4
**Confidence:** 3

**Summary:**

The paper introduces "Collapsed Effective Operators" to marginalize higher-order topological structures into a single vertex-level operator. The paper achieves this by lifting rank-specific operators into a Graded Laplacian and applying the schur complement to collapse the higher-order dynamics back onto the rank-0 (node) set. This methodology provides mathematical guarantees for PSDness. The operator is emperically evaluated on tasks like  manifold denoising, unsupervised protein segmentation, spectral clustering on SBMs and real-world networks.

**Compliance With Llm Reviewing Policy:**

Affirmed.

**Final Justification:**

The rebuttal addressed my own concerns, however some of the other reviews pointed out (quite severe) errors in notations and definition, so I do not feel comfortable raising my score from weak accept to accept.

**Key Questions For Authors:**

# Questions
**Q1** Given the discussion in remark 3.9, is there a tradeoff in the loss of the spectral gap? Are there datasets where we the loss of the spectral gap loses all performance - or does this not happen in practice? One potentially interesting experiment would be to run an experiment on SBM clustering, where you vary $q$ to be closer and closer to $p$, and compare standard SC to your approach.

**Q2** For some of the real-world normalized cut results the performance jumps are relatively small, e.g. a +3% bump on College Football and +2% on Les Misérables. Can you maybe elaborate on this? Is it always worth the additional overhead to use the topological machinery? Or are there cases where existing approaches are sufficient? Can we detect that before running our algorithms?

**Limitations:**

yes

**Strengths And Weaknesses:**

# Strengths
**S1** The ideas and theory introduced are quite neat. The operator is worked out and explained well, and the  and intuitions sense to me. My background is in spectral clustering so the proof that this operator mathematically reduces the effective resistance between nodes that share higher-order structures makes sense.

**S2** Experiment section is well done, and does show the effectiveness of the new operator.

# Weaknesses
**W1** Can you please expand a bit on computational complexity of your result? For the spectral clustering comparisons for example I’m curious whether your approach is scalable, since spectral clustering can scale to high number of vertices/edges as runtime is known to be nearly-linear in the number of edges. When taking a schur complement a matrix loses its sparsity and becomes dense, which leads to higher running times. I do see that the authors provide an implicit iterative solver (Algorithm 1) with Tikhonov regularization to avoid materializing the dense matrix.

# minor comments
Line 267 - should “small spectral gaps” be large spectral gaps here? Since you talk about the (normalized) laplacian?

---

> ### Author Rebuttal · Authors · 2026-03-31
>
> We thank the reviewer for their constructive feedback. Given their background in spectral clustering, we are especially encouraged that they found our derivations intuitive and our experiments compelling. In the following, we address their questions and concerns point by point.
>
> **W1 Missing computational complexity:**
>
> In applications such as spectral clustering, the dense collapsed operator $S_{\epsilon}$ does not need to be constructed explicitly. Instead, we employ a Krylov subspace approach which only requires matrix-vector multiplications of the form $v_{next} = S_{\epsilon} v$, i.e., $S_{\epsilon} v = A v - B(C + \epsilon I)^{-1}B^\top v$. Evaluating this product only requires solving $(C + \epsilon I) z = y$ via Conjugate Gradients.
>
> A single matrix-vector multiplication with the sparse block $C$ takes $\mathcal{O}(M)$ operations, where $M$ is the total number of non-zero entries (incidences) in the higher-order complex. Therefore, the runtime scales linearly in $\mathcal{O}(N_{CG} \cdot M)$.
> We conduct additional experimental analysis to support this:
>
> **Table: Runtime scaling with increasing higher-order topological density ($V=400$)**
> | Vertices | Edges | Triangles | Tetrahedra | Implicit Sparse CG |
> | :--- | :--- | :--- | :--- | :--- |
> | 400 | 696 | 18 | 0 | 0.50s |
> | 400 | 1650 | 296 | 4 |  12.43s |
> | 400 | 2081 | 625 | 12 |  36.32s |
> | 400 | 2702 | 1416 | 71 |  95.73s |
> | 400 | 4159 | 5504 | 1148 |  223.93s |
>
> It is worth noting that the Conjugate Gradient (CG) method is sensitive to matrix conditioning. Consequently, higher-order incidences may require more iterations if they worsen the operator's conditioning. Increased density mainly affects the cost per iteration through more expensive matrix-vector products. Nevertheless, when the regularized system retains a favorable sparse structure, CG remains practical and preferable to direct solvers for large-scale problems.
>
> **Runtime scaling in the sparse regime**
>
> | Vertices  | Edges | Triangles | Tetrahedra | Sparse Direct Factorization | Implicit Sparse CG (Ours) |
> | :--- | :--- | :--- | :--- | :--- | :--- |
> | 200 | 742 | 240 | 9 | 0.08s | 5.41s |
> | 800 | 4434 | 741 | 2 | 4.58s | 20.87s |
> | 1200 | 7311 | 939 | 7 | 33.59s | 43.96s |
> | 2000 | 13840 | 1180 | 1 | 323.14s | 75.51s |
> | 3000 | 23297 | 1571 | 1 | OOM | 404.77s |
>
> We observe that at $V=2000$, our implicit CG implementation already completes the operation more than 4x faster than the direct sparse solver (75.5s vs 323.1s).
>
> **Minor Comment (regarding line 267).** We thank the reviewer for flagging this sentence. The intended statement was that small spectral gaps correspond to sparse cuts / weak connectivity, consistent with Cheeger-type inequalities. We have revised the wording to be more precise.
>
> **Q1 clustering behaviour with different p,q.**
>
> | q/p | Std SC | Schur   | Intra-Δ % |
> | :--- | :--- | :--- | :---  |
> | 0.050 | 0.794  | 0.817   | 97.7 |
> | 0.100 | 0.735 | 0.762   | 90.7 |
> | 0.500 | 0.454  | 0.393    | 23.3 |
> | 0.875 | 0.355  | 0.332   | 7.8 |
>
> Collapsed-operator clustering outperforms graph-Laplacian clustering when $q/p \leq 0.10$. We observe that in this regime, lifted higher-order cells appear informative for $L_\star$, but the graph Laplacian can't exploit them. As q/p increases, this higher-order signal appears to become less discriminative. Nevertheless, we find it very interesting that a "filling-in" of some triangles and tetrahedra in these graphs yields an informative downstream signal, mediated by the collapsed operator $L^\star$.
>
> **Q2 Real-world applicability and anticipation.**
> We agree that on datasets such as College Football (+3%) and Les Misérables (+2%), the gains are more modest than on synthetic benchmarks such as SBM. These networks are dominated by pairwise homophily. In our analysis, adjacent nodes are approximately 4–8× more likely to share a label than a uniformly sampled node pair. This suggests that the graph Laplacian already captures most of the community structure, while higher-order cells mainly refine cluster boundaries rather than reveal entirely new structures, which explains the smaller but still consistent improvements.
>
> However, our collapsed operator integrates higher-order information in a single preprocessing step, so the overhead during clustering or training is modest compared to iterative higher-order message-passing models. Thus, even when gains are incremental, the cost can remain manageable.
>
> Whether the added complexity is worthwhile can often be assessed beforehand. If the data contains meaningful higher-order interactions, such as secondary/tertiary protein structure, higher-order information is likely beneficial. In other settings, identifying such structures may require domain knowledge, which we view as an interesting direction for future work.

---

> > ### Author Rebuttal · Reviewer_MmFQ · 2026-04-03
> >
> > I thank the authors for their thorough rebuttal. Having read the rebuttal and the other reviews I maintain my score.

---

### Decision · Program_Chairs · 2026-04-30

**Decision:**

Accept (regular)

**Comment:**

The paper introduces Collapsed Effective Operators, and yields a dense operator that encodes long-range interactions mediated by topology. The paper further shows the proposed approach performs better than spectral clustering experimentally. The reviewers pointed out several mistakes/issues on the mathematical analysis, and these issues were addressed during the reviewer-author discussion phase. The authors need to take these discussions into account and improve the presentation of their mathematical analysis when preparing the next version.